# Neural signatures of hyperdirect pathway activity in Parkinson's disease

Ashwini Oswal [1,2,3✉], Chunyan Cao[4], Chien-Hung Yeh[1,2,5], Wolf-Julian Neumann[6], James Gratwicke[7], Harith Akram[7], Andreas Horn [6], Dianyou Li[4], Shikun Zhan[4], Chao Zhang[4], Qiang Wang[6], Ludvic Zrinzo [7], Tom Foltynie[7], Patricia Limousin [7], Rafal Bogacz[1,2], Bomin Sun[4], Masud Husain [2], Peter Brown [1,2,8✉] & Vladimir Litvak [3,8✉]

Parkinson's disease (PD) is characterised by the emergence of beta frequency oscillatory synchronisation across the cortico-basal-ganglia circuit. The relationship between the anatomy of this circuit and oscillatory synchronisation within it remains unclear. We address this by combining recordings from human subthalamic nucleus (STN) and internal globus pallidus (GPi) with magnetoencephalography, tractography and computational modelling. Coherence between supplementary motor area and STN within the high (21–30 Hz) but not low (13-21 Hz) beta frequency range correlated with 'hyperdirect pathway' fibre densities between these structures. Furthermore, supplementary motor area activity drove STN activity selectively at high beta frequencies suggesting that high beta frequencies propagate from the cortex to the basal ganglia via the hyperdirect pathway. Computational modelling revealed that exaggerated high beta hyperdirect pathway activity can provoke the generation of widespread pathological synchrony at lower beta frequencies. These findings suggest a spectral signature and a pathophysiological role for the hyperdirect pathway in PD.

[1] MRC Brain Network Dynamics Unit, University of Oxford, Oxford, UK. [2] Nuffield Department of Clinical Neurosciences, University of Oxford, Oxford, UK. [3] The Wellcome Centre for Human Neuroimaging, University College London, London, UK. [4] Department of Neurosurgery, Affiliated Ruijin Hospital, School of Medicine, Shanghai JiaoTong University, Shanghai, China. [5] School of Information and Electronics Engineering, Beijing Institute of Technology, Beijing, China. [6] Department of Neurology, Charité University, Berlin, Germany. [7] Department of Clinical and Movement Neurosciences, University College London, London, UK. [8] These authors contributed equally: Peter Brown, Vladimir Litvak. ✉email: ashwini.oswal@ndcn.ox.ac.uk; peter.brown@ndcn.ox.ac.uk; v.litvak@ucl.ac.uk

Parkinson's disease is a common disorder of movement, which is characterized by nigrostriatal dopamine depletion and the emergence of stereotyped patterns of oscillatory synchronization within cortico-basal-ganglia circuits. Excessive synchronization across beta-band frequencies (13–30 Hz) is a hallmark of the Parkinsonian dopamine-depleted state. By simultaneously recording cortical activity with EEG or magnetoencephalography (MEG) and intracranial local field potentials (LFP) in patients undergoing surgery for the insertion of Deep Brain Stimulation (DBS) electrodes it is possible to explore patterns of long-range synchronization that emerge within cortico-basal-ganglia circuits[1–3]. Using this approach, it has been previously shown that the STN couples with motor/premotor activity at beta frequencies, with the cortex, predominantly driving STN activity[2,4].

In contemporary models of the basal-ganglia-thalamocortical loop, cortical activity is thought to be transmitted to subcortical regions by three streams—the hyperdirect, direct and indirect pathways—which act in conjunction to shape the dynamics of action initiation and selection. The direct and indirect pathways provide cortical inputs to basal ganglia via the striatum[5–8]. The hyperdirect pathway is a monosynaptic axonal connection, thought to be at least partly formed from axon collaterals of corticobulbar and corticospinal fibers, which runs from the frontal cortex to the STN and is proposed to provide rapid inhibition for action suppression[9–11]. The physiological properties of the hyperdirect pathway have been studied in humans and animals using a combination of techniques including tracer studies[12], evaluation of evoked responses to DBS[11,13,14], and non-invasive tractography[10,15,16].

An improved understanding of the relationship between cortico-basal-ganglia anatomical projections and the generation of beta-band oscillatory synchrony is essential to fulfilling a critical gap in our understanding of network dysfunction in PD and could inform the development of more spatially and temporally patterned DBS therapies[17]. In this regard, previous studies have speculated on the potential importance of an exaggerated hyperdirect pathway[4,18–21], but details regarding pathophysiological mechanisms are lacking.

Synchrony at low beta frequencies (13–21 Hz) is detectable in the parkinsonian STN and is considered to be pathological as it is suppressed by both DBS and L-dopa therapy[22–24] with some studies also demonstrating a correlation between the extent of treatment-related beta suppression and clinical improvement[4,25–28]. Previous work also demonstrates that synchronous activity between the cortex and the STN predominates

at higher beta frequencies (21–30 Hz) and is segregated such that mesial motor/premotor areas drive STN activity across this frequency range[2,4]. This leads to the hypothesis that cortical coupling with the STN at high beta frequencies may reflect hyperdirect pathway activity[4]. If high beta frequencies are reflective of the hyperdirect pathway, synchrony within this frequency range might be expected to be greater within the STN and its cortical network than within other basal ganglia structures such as the GPi. However, due to the connectivity between STN and GPi, any differences between these two sites are likely to be relative rather than absolute. Thus, we note that high beta activity has been detected in both the STN and the GPi[3,29]. In addition to the relative differences in power, we would predict an overlap between cortico-STN anatomical connectivity of the hyperdirect pathway and the profile of cortico-STN functional connectivity at high beta frequencies.

To test these hypotheses, we performed simultaneous MEG and intracranial LFP recordings in Parkinsonian patients undergoing surgery for the insertion of DBS electrodes in either the STN or the internal segment of the globus pallidus (GPi). A total of 32 patients were recruited from two separate surgical centers—one in London and one in Shanghai—and synchrony profiles of the STN and GPi networks were compared. Cortico-STN anatomical connectivity derived from individual patient electrode localizations and both individual and open-source tractography connectomes was integrated with individual patient MEG and LFP derived cortico-STN functional connectivity in order to establish the relationship between hyperdirect pathway fiber density and cortico-STN coupling at high beta frequencies.

Our findings indicate that cortical connectivity with the STN at high beta frequencies reflects activity within the hyperdirect pathway. Our empirical findings are recapitulated by a biophysical model which additionally reveals that an exaggerated hyperdirect pathway in PD may lead to the generation of subcortical synchrony at lower beta frequencies (13–21 Hz) which are considered to be more directly pathological.

## Results

**Differences in local synchrony between the STN and GPi and medication effects on the STN at beta frequencies.** Figure 1 shows trajectories and contact locations for electrodes targeting the STN (dark blue for UCL and turquoise for Shanghai) and GPi (white for UCL and yellow for Shanghai) in Montreal Neurological Institute (MNI) space (corresponding to STN UCL and Shanghai patients 1–6 and GPi UCL and Shanghai patients 1–6; see Supplementary Table 1). Contact localization for individual patients is shown in Supplementary Figs. 1 and 2. Contacts traversing either the STN or the GPi are colored in blue and only data from adjacent contact pairs where at least one contact traversed either the STN or GPi was used for subsequent analysis. For example, in the case that only the most inferior contacts 0 and 1 traversed the STN bipolar contact pairs 01 and 12 were used for subsequent analysis.

Figure 2A depicts the mean spectral power for all STN and GPi bipolar contacts separately for the UCL and Shanghai cohorts, whilst Fig. 2B reveals the mean spectral power of only the oscillatory component (after removing the aperiodic 1/f component) for the same STN and GPi contacts. In both structures, there is a peak centered at or below 10 Hz. Subjects also displayed oscillatory peaks within the low (13–21 Hz) and high beta (21–30 Hz) frequency ranges. For both the Shanghai and UCL cohorts, we observed that power within the high beta frequency range was significantly greater within the STN than within the GPi (gray lines in Fig. 2B; UCL: peak $t = 12.30$, FWE $p < 1 \times 10^{-3}$; Shanghai: peak $t = 4.32$, FWE $p = 2 \times 10^{-3}$). In addition, for the

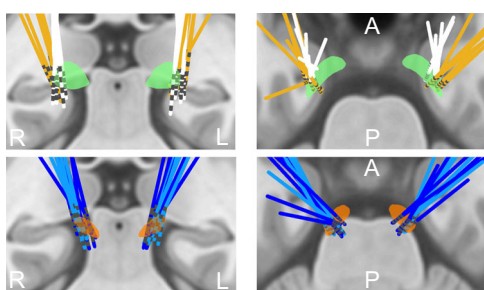

**Fig. 1 Localization of electrodes in MNI space.** The upper two images show electrodes targeting the GPi (green); electrodes from the Shanghai cohort are colored in yellow, whilst electrodes from the UCL cohort are colored in white. The bottom two images show electrodes targeting the STN (orange); electrodes from the UCL cohort are colored in dark blue whilst electrodes from the Shanghai cohort are colored in turquoise. Coronal (left) and axial (right) views are displayed and superimposed on a T1-weighted structural MRI. A—Anterior, P—Posterior, R—Right, L—Left.

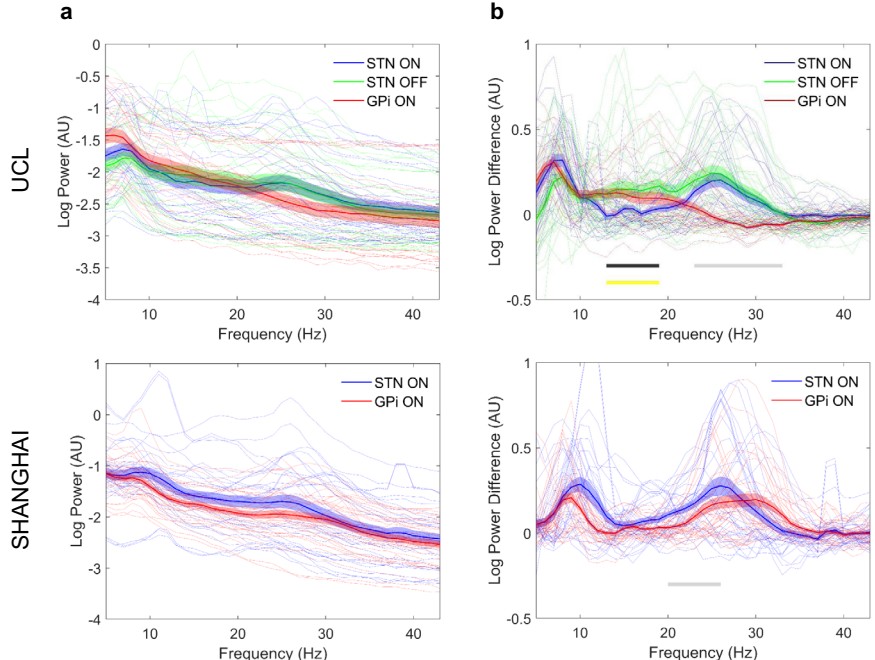

**Fig. 2 Local synchrony distinguishes the STN and GPi.** Mean log spectral power for all STN (blue line—ON medication STN recording; green line—OFF medication STN recording only performed at UCL) and GPi (red line) contacts are plotted separately for the UCL (upper image) and Shanghai (lower image) cohorts in Panel **A**. Panel **B** displays the mean log power of the oscillatory component after subtraction of the aperiodic 1/f component. The shaded regions indicate standard errors of the mean. Spectral profiles of individual STN and GPi contacts are also shown in both panels by the feint blue, green, and red lines. In both structures, there are oscillatory peaks below 10 Hz. Subjects also displayed oscillatory peaks within the low (13–21 Hz) and high beta (21–30 Hz) frequency ranges. The gray line in panel **B** indicates regions where oscillatory power was significantly greater in the STN than in the GPi (between 23–33 Hz for the UCL cohort and between 20–26 Hz for the Shanghai cohort) in the ON medication state. In the upper image of Panel **B** the black line indicates regions where spectral power in the GPi exceeded spectral power in the STN in the ON medication state (13–19 Hz). Finally, also in this image, the yellow line indicates regions for the STN where power in the OFF medication state was greater than power in the ON medication state (13–19 Hz).

UCL, but not for the Shanghai cohort we observed that low beta power was greater within the GPi compared to the STN (black line in Fig. 2B; peak $t = 6.58$, FWE $p < 1 \times 10^{-3}$). For the Shanghai STN and GPi cohort and for the UCL GPi cohort recordings were only performed ON medication. In contrast, for the UCL STN cohort in which patients underwent both ON and OFF medication recordings, we observed a region within low beta frequencies (yellow line in Fig. 2B; peak $t = 4.24$, FWE $p = 3 \times 10^{-3}$) where low beta power was increased in the OFF state compared to the ON state. Importantly there were no medication effects on high beta power within the STN.

**Cortical-subcortical coherence at high beta frequencies occurs preferentially within the cortico-STN network.** Comparison of the profile of cortical coherence for the STN and GPi for the UCL cohort, revealed an interaction between frequency band and electrode location such that specific cortical regions displayed greater coherence with the STN than they did with the GPi, preferentially at high rather than at low beta frequencies. Regions displaying this interaction included the SMA and mesial areas of the primary motor cortex (Fig. 3A upper panel; peak $t = 4.52$, FWE $p = 2 \times 10^{-3}$ at MNI coordinates 10 -46 82). In Fig. 3A the blue and green contour lines represent the boundaries of the primary motor cortex and the SMA derived from the Automated Anatomical Labelling (AAL) atlas. Interestingly the interaction effect did not extend as far laterally as the hand area of the primary motor cortex (see lower panel of Fig. 3A). In contrast, there was no main effect of the frequency band or electrode location (peak $t = 1.7$ and 1.9, respectively, in both cases FWE, $p > 0.1$).

In addition, we observed a simple main effect of the frequency band for the STN. Figure 3B depicts two mesial clusters that include SMA and the leg area of M1 (posterior cluster peak $t = 4.13$, FWE $p = 6 \times 10^{-3}$, at MNI coordinates 2 -48 72; anterior cluster peak $t = 4.44$, FWE $p = 2 \times 10^{-3}$ at MNI coordinates 20 -14 72), where the cortico-STN coupling is greater at high rather than at low beta frequencies. Finally for the STN UCL cohort comparison of cortico-STN coherence profiles ON and OFF medication revealed only a main effect of the band, such that mesial motor areas including SMA were preferentially coupled to the STN at high rather than at low beta frequencies (Fig. 3C; peak $t = 5.24$, FWE $p = 4 \times 10^{-3}$, at MNI coordinates 16 12 70). We observed no significant main effect or interaction of medication state.

To further visualize these effects, and provide an indication of effect sizes, source extracted coherence spectra are shown in Fig. 3D–E. Figures 3D and 3E show coherence profiles of the STN and GPi for the peak locations within the mesial anterior (SMA) and mesial posterior (mesial primary motor cortex, M1) clusters for which there was a simple main effect of the band for the STN (visualized in Fig. 3B). The cortical coherence profiles of the STN reveal distinct peaks within the low (~13 Hz) and high (~27 Hz) beta frequency ranges. Cortical coherence at high beta frequencies is seen to be greater for the STN than for the GPi.

Corresponding results from the Shanghai cohort are displayed in Fig. 4. In this cohort, there were significant main effects of both band and site. Figure 4A displays a mesial cluster encompassing SMA for which there was a main effect of band such that coherence at high beta frequencies was greater than that at low beta frequencies for both STN and GPi (peak $t$ statistic = 5.09,

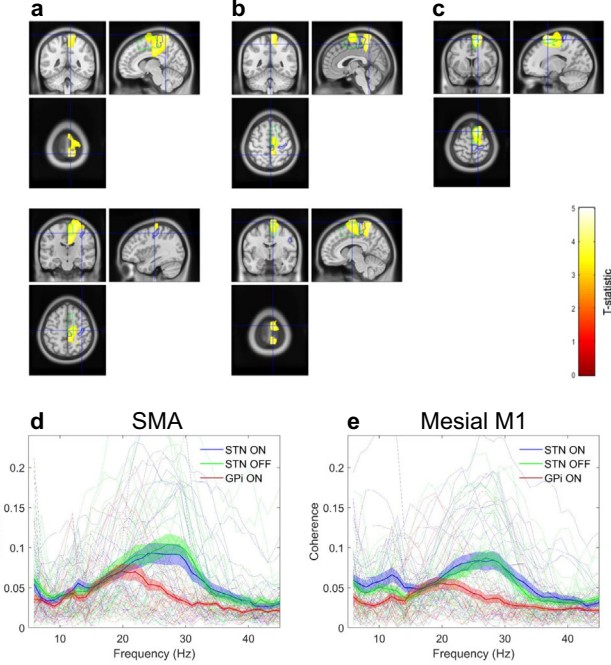

**Fig. 3 Differences in cortico-STN and cortico-GPi coherence for the UCL cohort; high beta band coherence segregates to the cortico-STN network and is largely uninfluenced by dopaminergic medication.** Panels **A** and **B** are SPMs showing results of the 2x2 ANOVA with factors frequency (low beta versus high beta) and target nucleus (STN versus GPi). *T*-values of voxels within significant clusters are superimposed onto a T1-weighted MRI with the color bar indicating the value of the t-statistic. Contours of the SMA and primary motor cortex are shown in green and blue, respectively. **A** The upper panel displays regions where there was a significant interaction, such that there was greater cortical coherence with the STN than with the GPi at high rather than at low beta frequencies. The cluster encompasses the mesial primary motor cortex and the SMA. Cross-hairs are centered on the location of the peak t-statistic at 10 -46 82. In the lower panel, the cross-hairs are centered at the location of the hand area of the primary motor cortex, highlighting that the interaction effect was centered medially. **B** Two clusters displaying a significant simple main effect of the band for the STN. The clusters include SMA and the mesial primary motor cortex. In the upper panel, the cross-hairs are centered on the location of the peak t-statistic within the posterior cluster at MNI coordinates 2 -48 72. In the lower panel cross-hairs are centered on the location of the peak t-statistic within the anterior cluster at MNI coordinates 20 -14 72. Panels **C** shows results of the 2 × 2 ANOVA with factors frequency (low beta versus high beta) and medication state (ON versus OFF) for the STN UCL cohort. There was a significant main effect of frequency such that the SMA exhibited higher coherence with the STN at high rather than at low beta frequencies (peak t-statistic at MNI coordinates 16 12 70). **D** and **E** show group mean (with standard errors) and individual coherence spectra computed between the STN/GPi and cortical locations of the peak t-statistic of the simple main effect of the band for the STN (these correspond to the anterior and posterior clusters in **C** which are located in the SMA and in the mesial primary motor cortex (M1)).

FWE $p < 1 \times 10^{-3}$, at MNI, coordinates 2 -2 76). Similarly, Fig. 4B displays a cluster centered on mesial primary motor cortex for which there was a main effect of site, such that beta coherence summed across the low and high sub-bands was greater for the STN than for the GPi (peak $t$ statistic = 3.91, FWE $p = 0.01$, at MNI coordinates 8 -52 82). Importantly, although a significant interaction between site and frequency band was not observed for the Shanghai cohort, a site-specific simple main effect of frequency band was seen—such that high beta coherence was

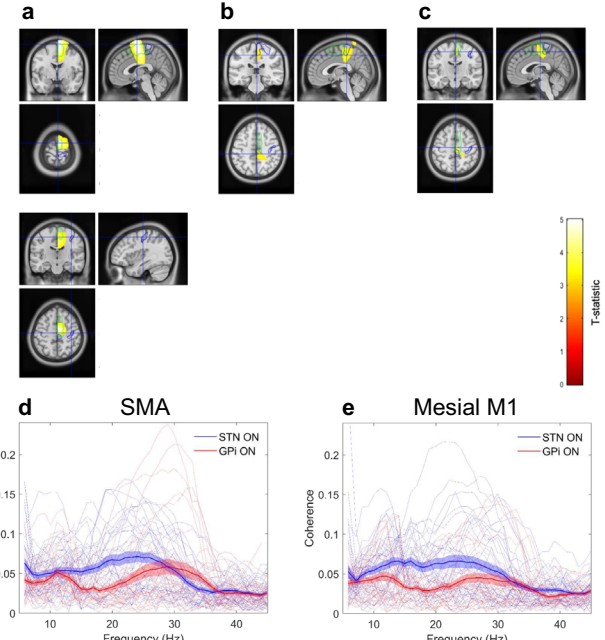

**Fig. 4 Differences in cortico-STN and cortico-GPi coherence for the Shanghai cohort; high beta band coherence segregates to the cortico-STN network.** As per Fig. 2 panels **A**–**C** are SPMs showing results of the 2 × 2 ANOVA with factors frequency (low beta versus high beta) and target nucleus (STN versus GPi). **A** The upper panel displays regions where there was a significant main effect of the band, such that cortical coherence with both the STN and GPi was greater at high rather than at low beta frequencies. The cluster encompasses the SMA. Cross-hairs are centered on the location of the peak t-statistic at 2 -2 76. In the lower panel, the cross-hairs are centered at the location of the hand area of the primary motor cortex, highlighting that the main effect was centered medially. **B** A cluster displaying a significant main effect of site, such that coherence across the high and low beta bands was greater for the STN than for the GPi, is shown. The cluster includes the mesial primary motor cortex and the cross-hairs are centered on the location of the peak t-statistic at MNI coordinates 8 -52 82. **C** A cluster lying within the SMA, for which there was a simple main effect of the band for the STN is displayed. The cross-hairs are centered on the peak t-statistic location at MNI coordinates 2 -16 54. Panels **D** and **E** show group mean (with standard errors) and individual coherence spectra computed between the STN/GPi and cortical locations of the peak t-statistic separately for the main effect of the band (**D**) and site (**E**). These correspond to the clusters in **A** and **B** which are located in the SMA and in the mesial primary motor cortex (M1).

greater than low beta coherence with the SMA for the STN, but not for the GPi (Fig. 4C; peak $t$ statistic = 4.05, FWE $p = 7 \times 10^{-3}$, at MNI coordinates 2 -16 54). Figures 4D and 4E show coherence profiles of the STN and GPi for the peak locations within the SMA and mesial primary motor cortex for which there were main effects of band and site, respectively, (visualized in Fig. 4A and B). As per the UCL cohort, the cortical coherence profiles of both the STN and GPi reveal distinct peaks within the low and high beta frequency ranges. Cortical coherence at high beta frequencies is seen to be greater for the STN than for the GPi.

In summary, the findings from both surgical centers are consistent and highlight the segregation of coherence at high beta frequencies to the SMA/M1—STN network.

**Directionality of beta-band cortical-subcortical coupling and estimation of transmission delays.** Based on the above evidence

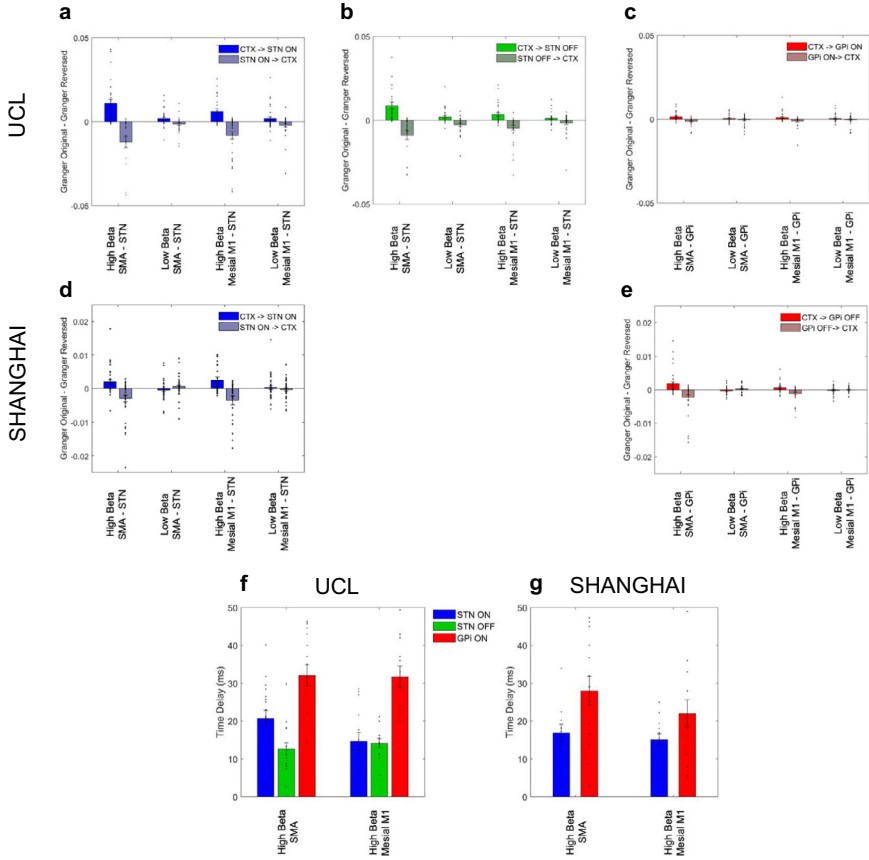

**Fig. 5 Granger directionality and analysis of net time delays: cortical driving of STN and GPi at high beta frequencies.** Group means differences in Granger causality between the original data and time-reversed data are averaged across the high (21–30 Hz) and low (13–20 Hz) beta frequency ranges for the SMA-STN and mesial M1-STN networks in the ON medication condition (**A** UCL cohort and **D** Shanghai cohort). Panel **B** shows results from the UCL cohort in **A** in the OFF medication condition. Panels **C** and **E** show results for the SMA-GPi and mesial M1-GPi networks from both surgical centers in the ON medication condition. Source time series for Granger causality computation are extracted from the locations of peak $t$ statistics of main effects separately for the STN and the GPi as per Figs. 3D, 3E, 4D, and 4E. The difference in Granger causality is significantly greater than zero in the direction of SMA and mesial M1 to both STN and GPi for the upper beta frequency band. For the same cortical areas that drive sub-cortical activity, Granger causality estimates are negative in the (reverse) direction of subcortical sites driving cortex, confirming that cortical activity leads the former. In the case of there being statistically significant unidirectional coupling, time delays between cortex and the STN/GPi were estimated (**F** and **G**). Time delays were estimated for the same cortical locations and frequency bands used for Granger causality analysis. Mean values, standard errors, and individual data points are shown. For both the UCL and Shanghai cohorts, $n = 6$ patients (see Supplementary Table 1 and Supplementary Figs. 1 and 2 for details of the electrodes and contact pairs that were included for each patient).

motor cortical coupling with the STN and GPi is divided into high and low beta frequencies. Coupling at high beta frequencies appears to be less for the GPi than for the STN. We characterized the directionality of cortico-subcortical coupling at high and low beta frequencies in order to determine whether these signals originate from the cortex or subcortical structures. For the purposes of this analysis, we used source locations derived from the locations of the peak $t$ statistics used for the computation of source extracted coherence profiles as described above.

For both the UCL and Shanghai cohorts, the difference in Granger causality was significantly greater than zero in the direction of SMA and mesial M1 leading the STN for high but not for low beta frequencies (Fig. 5AB and D). For the SMA-STN network one-sample $t$-tests for the high beta sub-band were; UCL ON medication: $t_{25} = 3.88$, $P < 1 \times 10^{-3}$, UCL OFF medication: $t_{25} = 3.60$, $P < 1 \times 10^{-3}$, Shanghai: $t_{29} = 2.72$, $p < 1 \times 10^{-3}$. In addition, for the mesial M1-STN network the statistics were as follows: UCL ON medication: $t_{25} = 3.19$, $P = 2 \times 10^{-3}$, UCL OFF medication: $t_{25} = 3.33$, $P = 1 \times 10^{-3}$, Shanghai: $t_{29} = 2.76$, $P < 1 \times 10^{-3}$.

Similarly cortical activity in SMA and mesial M1 led activity within the GPi selectively over the high but not low beta frequency range (Fig. 5C and E). For the SMA-GPi high beta network: UCL ON medication: $t_{20} = 2.5$, $P = 0.01$, Shanghai $t_{27} = 2.2$, $P = 0.02$. Similarly for the high beta mesial M1-GPi network: UCL ON medication: $t_{20} = 1.98$, $P = 0.03$, Shanghai $t_{25} = 2.7$, $P < 0.01$.

After directionality analysis, net time delays in the high beta band between the SMA and mesial M1 regions and the STN/GPi were estimated (Fig. 5F and G). Time delays were estimated for these networks as there was predominant unidirectional coupling in the direction of the cortex to both the GPi and the STN. Given that the GPi lies further downstream of the STN in the motor cortico-basal ganglia circuit and lacks hyperdirect pathway inputs, we tested whether time delays from cortex to STN were shorter than those from cortex to GPi. Accordingly, we set up a $2 \times 2$ ANOVA, adding covariates as previously described, with factors cortical location (SMA versus Mesial M1) and target location (STN versus GPi). Our results revealed a significant main effect of target location such that delays between SMA and mesial

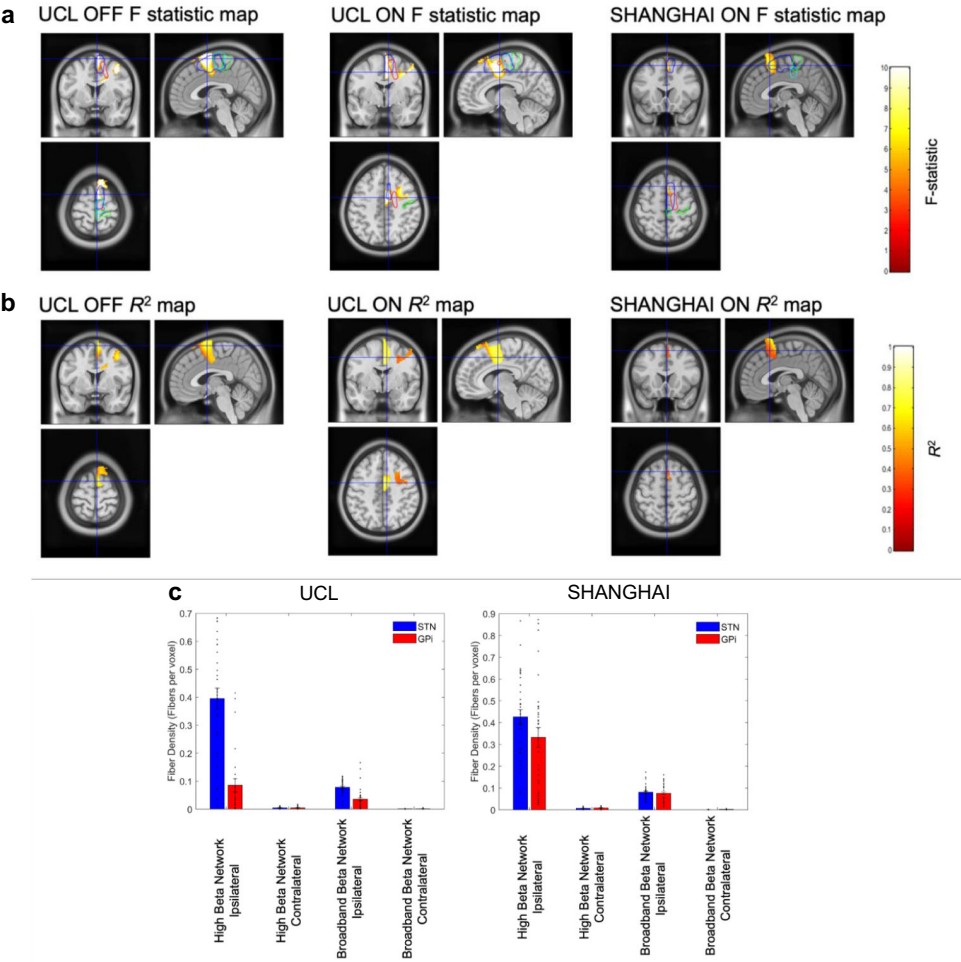

**Fig. 6 Relationship between connectome derived structural and individual patient functional connectivity: tract density predicts hyperdirect pathway high beta coherence. A** Statistical image of significant clusters for the group level voxel-wise correlation between high beta band cortico-STN coherence and cortico-STN fiber density separately for the UCL (ON and OFF medication) and Shanghai cohorts (ON medication). The images are superimposed on a T1-weighted MRI and the crosshairs are centered on the location of the peak F-statistic of the cluster, which in all cases lies within the SMA. The blue and green contours enclose the volumes bounded by the SMA and primary motor cortex (M1) derived from the AAL atlas. The red and turquoise contours indicate regions were at the group level high beta band coherence and tract density, respectively, were greater than the 95% percentile. **B** The $R^2$ correlation coefficient maps for voxels within each significant cluster are displayed. In all cases, there are significant clusters within the SMA. **C** Mean tract density estimates are plotted for the ipsilateral and contralateral 'high beta' and 'broadband beta' networks separately for STN (blue) and GPi (red) contacts for the two surgical centers. Vertical bars represent standard errors of the mean and individual data points are also shown. For both the UCL and Shanghai cohorts, $n = 6$ patients (see Supplementary Table 1 and Supplementary Figs. 1 and 2 for details of the electrodes and contact pairs that were included for each patient).

M1 to STN were shorter than those from the same cortical areas to the GPi (UCL: $F(1,52) = 59.8$, $P < 1 \times 10^{-3}$; Shanghai: $F(1,40) = 15.98$, $P < 1 \times 10^{-3}$) Importantly there was no significant main effect of cortical location (UCL: $F(1,52) = 3.05$, $P = 0.09$; Shanghai: $F(1,40) = 0.8$, $P = 0.38$), nor was there a significant interaction of the two factors (UCL: $F(1,52) = 3.10$, $P = 0.08$; Shanghai: $F(1,40) = 1.26$, $P = 0.27$). In a separate analysis, we confirmed a lack of medication effects on delays for the UCL cohort (UCL: $F(1,52) = 3.07$, $P = 0.09$).

**Functional connectivity is predicted by anatomical connectivity within the cortico-STN hyperdirect pathway.** Our core hypothesis is that activity within the hyperdirect pathway is indexed by cortico-STN coherence at high beta frequencies. Accordingly, we tested for a voxel-wise relationship between cortico-STN tract density and cortico-STN high beta band coherence across all studied contacts for each hemisphere and each subject. The results of cluster-based permutation testing are

shown in Fig. 6A, and reveal a lateralized cluster encompassing the SMA (blue contour), where tract density was predictive of high beta band coherence for the UCL dataset both ON and OFF medication and for the Shanghai dataset. $R^2$ correlation coefficient maps for voxels within each significant cluster are displayed in Fig. 6B. Further analysis revealed no significant relationship between cortico-STN tract density and cortico-STN low beta band coherence. A similar correlation of cortico-GPi tract density with cortico-GPi coherence in the high and low beta frequency bands was also not significant.

For further visualization of the overlap between MEG derived functional connectivity and tractography derived structural connectivity, we studied fiber tracts passing through the predefined spherical ROI for each contact which originated within cortical volumes for which: (1) there was either a main effect of the band (for the Shanghai cohort; see Fig. 4A) or a simple main effect of the band for the STN (for the UCL cohort; see Fig. 3B)—which we name the 'high beta network' and (2)

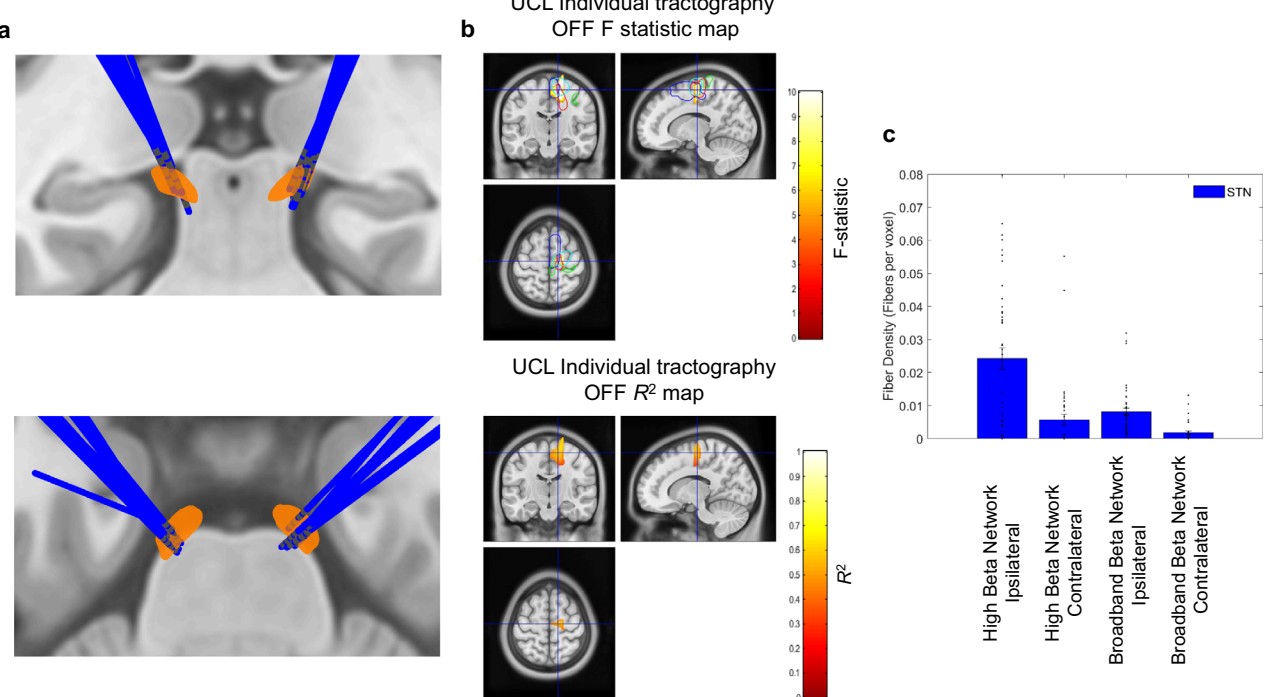

**Fig. 7 Relationship between individual patient structural and individual patient functional connectivity: tract density predicts hyperdirect pathway high beta coherence. A** Electrode localization for eight STN UCL patients (patients 7–14 in Supplementary Table 1) for whom individual subject structural and functional connectomes were available. **B** Statistical images of significant clusters for the group level voxel-wise correlation between high beta band cortico-STN coherence and cortico-STN fiber density as per Fig. 6. The crosshairs are centered on the location of the peak F-statistic of the cluster, which lies within the SMA (blue contour). A corresponding $R^2$ map is also shown for voxels within the cluster. **C** Mean tract density estimates are plotted for the ipsilateral and contralateral 'high beta' and 'broadband beta' networks. Vertical bars represent standard errors of the mean and individual data points are also shown. Data are shown for $n = 8$ patients (see Supplementary Table 1 and Supplementary Fig. 1 for details of electrodes and contacts that were included for each patient).

coherence with the STN/GPi across the entire beta frequency range (which we name the 'broadband beta network') was greater than coherence at alpha band frequencies. Group analyses are displayed in Supplementary Figs. 3 and 4. Tract densities were computed for the ipsilateral and contralateral high beta and broadband beta networks, separately for STN and GPi contacts. Results are summarized in Fig. 6C and reveal that STN contacts tend to have denser fiber innervations than GPi contacts from ipsilateral cortical regions that couple to them preferentially at high beta frequencies. We explored this phenomenon statistically by constructing a $2 \times 2 \times 2$ factorial ANOVA with factors network (high beta versus beta), laterality (ipsilateral versus contralateral), and electrode location (STN versus GPi)(see Supplementary Results for further details).

In a separate analysis, rather than limiting chosen fibers to those bypassing the striatum (indicative of hyperdirect connections), we selected fibers that passed through the striatum on their passage to either the STN or GPi. This procedure served to select fibers that may form part of the indirect and direct pathways to the STN and GPi, respectively. Using this approach we observed no significant relationship between tract densities and coherence in either the high or low beta bands for both cohorts. Taken together, our findings suggest that cortico-STN coherence in the high beta band is related strongly to structural connectivity within the hyperdirect pathway.

Finally, for the separate cohort of UCL patients with both individual structural connectomes and combined MEG-LFP recordings (UCL STN patients 7–14; see Supplementary Table 1), we observed an almost identical relationship between structural and functional connectivity within the hyperdirect pathway.

Electrode localization for this cohort is displayed in Fig. 7A, whilst Fig. 7B displays the corresponding F statistics and $R^2$ correlation maps. In keeping with the results obtained from PPMI connectome data, there is a lateralized cluster including the SMA where is tract density is predictive of high beta band coherence. Finally, Fig. 7C displays tract densities for STN connections to the 'high beta' and 'broadband beta networks'. In keeping with the results of the PPMI connectome, this highlights that the STN receives a high density of fibers from the ipsilateral 'high beta' network (see Supplementary Results for statistics).

**The computational model describing differential mechanisms by which high and low beta frequencies are generated in the cortico-basal-ganglia circuit.** We have provided empirical evidence that the hyperdirect pathway drives high beta activity in the basal ganglia. However, what is the origin of beta activity in the lower frequency band, and to what extent are the two activities functionally distinct[4,30,31]? It is after all the low beta activity that is believed to be pathological. To answer these questions, we turned to computational modeling and designed a model able to capture a number of empirical features observed both in our own data and in those of other electrophysiological studies. Work in computational neuroscience suggests that circuits composed from homogeneous populations of excitatory and inhibitory neurons may generate oscillations with a single dominant frequency[32]. To produce two frequencies in the model, we included two pairs of excitatory-inhibitory network generators: (1) an excitatory-inhibitory cortical network capable of generating high beta frequency oscillations and (2) the reciprocal excitatory-inhibitory

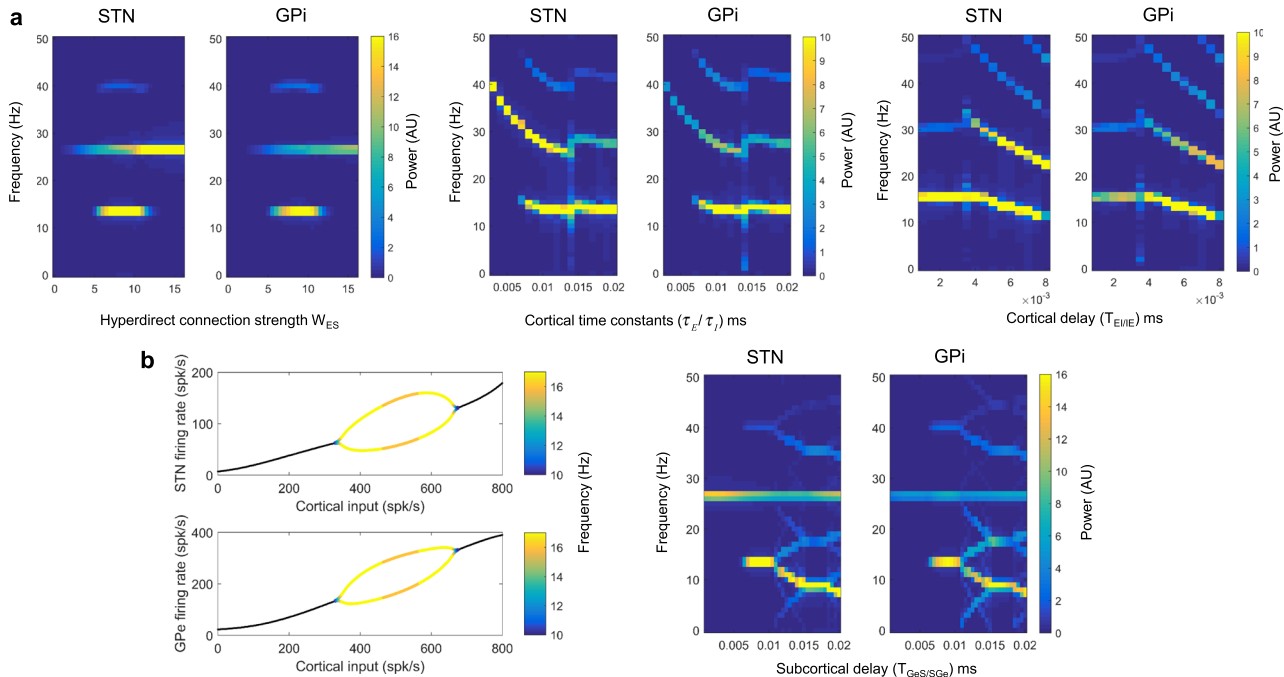

**Fig. 8 Computational modeling: the cortex generates high beta activity which is propagated subcortically and provokes the generation of pathological lower beta frequencies. A** We explore the effect of altering the hyperdirect connection strength ($W_{ES}$, left figure), cortical time constants ($\tau_E/\tau_I$, middle figure), and the cortical delay ($T_{IE}/T_{IE}$, right figure) on the spectra of the STN and GPi. High beta frequency activity generated within the cortex propagates to the STN and GPi. In addition, under certain parameter values, a lower beta frequency (between 10 and 15 Hz) emerges. The left image in **B** is a bifurcation diagram highlighting that low beta frequency oscillatory activity may be generated in the STN-GPe feedback loop depending on cortical inputs. In the oscillatory region of STN and GPe firing the dominant frequency is color-coded. The right image in **B** displays the effect on STN and GPi spectra of changing the subcortical delay parameters ($T_{GeS}$ /$T_{SGe}$).

STN-GPe subcortical network which served to generate low beta frequency oscillations[18,33].

The structure of our model and its ability to generate high beta activity cortically and low beta activity subcortically is illustrated in Supplementary Fig. 4.

**Modeling the effects of cortico-subcortical connectivity on the frequency and coherence of oscillations**. Figure 8 illustrates the effects of cortical input on the frequencies of oscillations within the STN-GPe loop. The left image in A shows the effect of modulating the hyperdirect pathway connection strength, $W_{ES}$ on the simulated spectra of the STN and GPi. As $W_{ES}$ is increased from 0, high beta oscillations from the cortex start to appear subcortically. At values of $W_{ES}$ between 5 and 10, low beta frequency oscillations (~13 Hz) appear in the spectra of STN and GPi and then disappear when $W_{ES}$ exceeds a value of 10. The middle and rightmost images in A explores the effects of varying cortical time constants ($\tau_E/\tau_I$) and the delay between the two cortical populations ($T_{EI/IE}$). Varying these cortical parameters primarily influences the frequency of the high beta oscillatory activity incoming from the cortex, with little effect on the low beta peak frequency.

Our simulations indicate how a strong hyperdirect pathway may lead to the generation of low beta frequency oscillations subcortically. To illustrate this more clearly, in the left image of Fig. 8B is a bifurcation diagram for STN and GPe activities for a reduced model of only the reciprocal STN-GPe network, where both populations receive striatal input and the STN receives fixed (non-oscillatory) excitatory cortical inputs. The x-axis plots the cortical input which is the product of the hyperdirect pathway connection strength, $W_{ES}$, and the firing rate of the cortical excitatory population. The system displays a stable fixed point (black lines)

which transitions to instability and oscillatory behavior at cortical input values between approximately 330–670 spikes/s. Above a spike range of approximately 670 spikes/s oscillatory behavior ceases as a stable fixed point returns. Within the oscillatory range, the peak of the low beta frequency is indicated in the color bar. Interestingly this simulation predicts that very high hyperdirect pathway strengths lead to the loss of low beta oscillations in the sub-cortex. The right image in Fig. 8B shows how the peak frequency of the low beta oscillation generated in the STN-GPe loop can be influenced by the transmission delays between these two structures ($T_{GeS/SGe}$).

Next, using our model we simulated power spectra of the cortex, STN, and GPi, and coherences between cortex-STN and cortex-GPi (Fig. 9A). Parameters used are listed in Supplementary Table 2, but we varied the value of the strength of the net inhibitory loop, $W_{GiE}$, between the GPi, thalamus, and cortex. In these simulations, the noise was added (see Supplementary Methods) in order to make coherence values physiologically plausible. A range of values of $W_{GIE}$ oscillatory peaks within the high and low beta frequency ranges are seen in the STN and GPi. As observed in our own data (Figs. 2, 3, and 4), both high beta band power and cortical coherence were greater for the STN than for the GPi. Increasing $W_{GiE}$ led to an increase in the amplitude of the low beta frequency peak in both the STN and the GPi in addition to increases in cortico-STN and cortico-GPi coherence at low beta frequencies. Interestingly, cortical activity predominantly displays a high beta frequency peak without the low beta frequency peak observed in subcortical activity. This is in keeping with electrocorticographic studies in PD patients which display spectral peaks above 20 Hz[34–36].

Finally, Fig. 9B reveals that the computational model developed predicts a monotonically increasing relationship between hyperdirect pathway strength $W_{ES}$ and cortico-STN coherence in the

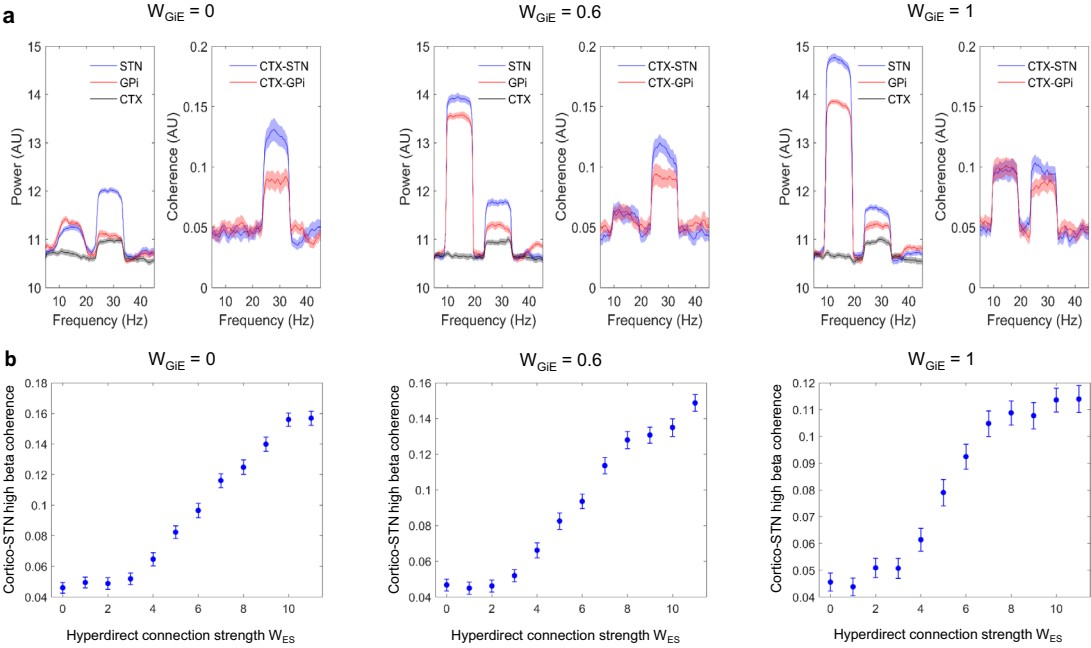

**Fig. 9 Computational modeling captures observed power and coherence spectra and predicts the observed relationship between hyperdirect pathway structural and functional connectivity. A** Simulated power spectra for the STN, GPi, and cortex are shown with the associated profiles of cortico-STN and cortico-GPi coherence spectra. Parameter values for this simulation are as per Supplementary Table 2, except for the fact that the strength of the net inhibitory loop, $W_{GIE}$, between the GPi, thalamus, and cortex was varied as shown. Power and coherence spectra display peaks at high and low beta frequencies, in keeping with experimental data. Increasing $W_{GIE}$ results in increased low beta frequency power and coherence. The shaded areas represent standard errors of the mean. **B** The simulated relationship between hyperdirect pathway strength and cortico-STN high beta band coherence displays a monotonically increasing profile. Standard errors of the mean are computed over $n = 50$ simulations.

high beta frequency range (21–30 Hz). The slope of this relationship is greatest above values of $W_{ES}$ that lead to the propagation of high beta oscillations sub-cortically (see Fig. 8A).

In summary, modeling reveals that the hyperdirect pathway propagates high beta frequencies subcortically and may contribute to the generation of subcortical low beta frequency rhythms. In addition, connections from basal ganglia via the thalamus to cortex may underlie the coherence between GPi/STN and cortex at low beta frequencies. The model also captures the empirical relationship between structural and functional connectivity within the hyperdirect pathway.

## Discussion

This study characterized the intersection between structural and functional connectivity within the basal ganglia in PD patients undergoing DBS surgery. We were able to isolate electrophysiological markers of hyperdirect pathway activity by integrating high-resolution dMRI derived structural connectivity—from both connectomes and individual patient dMRI—with functional connectivity derived from simultaneous MEG and STN LFP recordings. In addition, we compared the power and cortical coherence profiles of the STN and GPi, which revealed features suggestive of a relative lack of hyperdirect inputs to the latter. In comparison to GPi activity, STN activity displayed both greater local synchrony, as indexed by LFP amplitude, and greater functional connectivity with motor cortical areas, including the SMA and mesial primary motor cortex, at high beta frequencies. Furthermore, these same cortical regions tended to drive STN activity within the high beta frequency range with shorter delays than those that were observed for coupling to the GPi. More strikingly we show across participants that cortico-STN coherence at high beta frequencies correlates with hyperdirect pathway

fiber tract densities in a focal premotor region encompassing the SMA—an area that plays a key role in the volitional control of movement[37]. Collectively these data provide evidence that hyperdirect pathway activity within the cortico-basal-ganglia circuit evokes a unique spectral signature at high beta frequencies.

We interpret the data in the context of a computational model which provides important insight into the origins of beta oscillatory activity in PD. Our model demonstrates how an exaggerated hyperdirect pathway in PD is capable of both generating high beta frequency oscillations and inducing synchrony at lower beta frequencies within subcortical circuits. The model, therefore, provides a formal biophysical mechanism for frequency transduction in cortico-basal ganglia circuits, which hitherto has only been speculated[4,31]. Excessive sub-cortical synchrony at low beta frequencies is considered to be closely related to motor impairment in PD[38] and therefore strategies aimed at modulating the hyperdirect pathway may exert therapeutic effect by reducing subcortical transduction to pathological low beta activity.

**The intersection between structural and functional connectivity within the cortico-basal-ganglia circuit.** We report a correlation between cortico-STN coherence and cortico-STN fiber tract density that was specific anatomically to mesial premotor areas (SMA) and spectrally to upper beta-band frequencies. Importantly this correlation was not observed in the profile of cortical coherence with the GPi. We interpret this finding as being indicative of a tight relationship between structural and functional connectivity within the monosynaptic hyperdirect pathway from the cortex to the STN[9,15,39]. Importantly we isolated hyperdirect pathway fibers by studying fibers bypassing the striatum. When only striatal traversing fibers were considered, which may be reflective of indirect and direct

pathway connections to the STN and GPi, no relationships between structural and functional connectivity were observed.

Of particular relevance to our findings is the strong theoretical relationship between functional and effective—and implicitly also structural—connectivity in networks such as the hyperdirect pathway with unidirectional information transfer[40]. Previous studies examining the intersection of tractography and fMRI are suggestive of a similar overlap of structural and functional connectivity within hyperdirect connections during stopping behavior[10,41] with the strength of hyperdirect connections also correlating with the efficacy of stopping[42]. More broadly, the relationship between functional and structural connectivity in brain networks is highly complex[43] and also limited by the pitfalls of the modalities used to assess these measures. Nevertheless, significant overlaps between structural and functional connectivity have been noted in highly conserved networks such as the default mode network[44].

We hypothesize that the lack of an observed correlation between structural and functional connectivity within the cortico-GPi network is indicative of the fact that the GPi receives less synchronized cortical inputs through spatiotemporal intermixing of hyperdirect, direct and indirect pathway inputs that are themselves mediated by other nuclei before reaching the GPi[45]. Interestingly however the existence of a direct connection between the cortex and the GPi has recently been postulated and this would explain our finding of fibers passing between these two structures which bypass the striatum[46]. Our data suggest that this pathway is likely to be less dominant than the hyperdirect pathway based on the comparison of tract densities.

We estimated a delay of ~15–20 ms for cortico-STN hyperdirect transmission in the high beta band. Previous studies of cortical evoked responses to STN stimulation have identified the monosynaptic hyperdirect pathway based on the observation of response latencies ranging from 2 to 8 ms[13,14,47,48]. Importantly evoked response latencies are likely to be shorter than delays computed from phase-based estimates for a number of reasons. Firstly, delays from phase-based estimates depend not only on conduction delays but also on synaptic integration delays (in other words the delay for a neuronal population to synchronize its own activity to that of an input) which increase as more synapses are involved and may be extended by inhibitory inputs to the STN. Secondly, it is possible that there will be some mixing of hyperdirect and indirect pathway components that overlap in frequency leading to an increased estimate of time delays between the cortex and the STN[4,49]. We found relatively little evidence of this in our own data however since we only observed a correlation between cortico-STN hyperdirect fiber density and cortico-STN high beta band coherence and not between cortico-striatal-STN fiber densities (indicative of indirect pathway connections) and cortico-STN high beta band coherence. These observations suggest that the hyperdirect pathway may be the predominant route of transmission of high beta activity to the STN.

The above reasons may explain discrepancies in hyperdirect pathway evoked response latencies (which were ~2 ms) and delays estimated from cross-correlation measures (which were ~60 ms) in a recent paper published by Chen and colleagues[11,50]. Importantly the magnitude of cross-correlation at these larger latencies predicted stopping behaviors, highlighting that in vivo information transmission within the hyperdirect pathway may occur significantly more slowly than evoked response latencies. Importantly we also observed relative differences in transmission delays from the cortex to the STN and GPi. The finding of longer delays to the GPi, which lies further downstream in the cortico-basal-ganglia circuit supports the notion that high beta activity originates within the cortex and is propagated to basal ganglia.

**Insights from the computational modeling.** The model presented here underscores the likely importance of an exaggerated hyperdirect pathway in PD, which has also been suggested in previous reports[4,18,21,39]. More specifically we show how the exaggerated subcortical propagation of high beta frequencies via the hyperdirect pathway can lead to the generation of the lower beta frequencies that are believed to play a more direct role in the genesis of motor impairment[38]. This explanation is consistent with our own and previous findings that the net directionality of high beta frequency oscillations is from cortex to STN[4,51]. In contrast, we observed no significant net directionality for low beta frequency oscillations, which may reflect low subcortico-cortical feedback strengths via the GPi-thalamo-cortical loop (parameter $W_{GiE}$ in Supplementary Table 2). Secondly, our model predicted a monotonically increasing relationship between hyperdirect pathway strength and cortico-STN high beta band coherence, which is in keeping with the observed tight relationship between structural and functional connectivity within the hyperdirect pathway. The model also captures the relative difference in local and cortical synchrony in the high beta band for the STN and GPi, in the context of strong hyperdirect pathway inputs.

Further in keeping with our model is the finding that DBS—which may exert therapeutic benefit by suppressing the hyperdirect pathway[4,14,21,52]—has the effect of reducing coherence in the upper beta frequency band between the cortex and the STN and also the cortex and the GPi, in addition to suppressing low beta synchrony locally within the STN and the GPi[4,36,53].

Finally, it is worth mentioning that although the data in this report are consistent with a role of an exaggerated hyperdirect pathway in the generation of abnormal synchrony in PD, other reports in monkeys and rodents suggest that a loss of hyperdirect pathway inputs to the STN can also reproduce the PD phenotype[54,55]. Interestingly, this finding can be reconciled with our model when examining Figs. 8A and 8B. If we assume that in the healthy state hyperdirect pathway strength is high, (e.g., $W_{ES} > 12$) this will correspond to the stable regime of the STN-GPe loop in Fig. 8B where high beta band activity is propagated from the cortex to the subcortex. Pathological decreases in hyperdirect pathway strength will result in the system approaching a bifurcation and then additionally producing low beta frequency oscillations.

**Medication effects on high and low beta band power and coherence.** For the UCL STN cohort, in keeping with previous reports, we observed no medication effects on coherence or on power within the high beta frequency range[2,30,56,57]. Levodopa administration did however lead to a significant reduction in low beta band power within the STN as previously described[22,27,30]. Importantly the relationship between hyperdirect pathway tract density and high beta band cortico-STN coherence was maintained regardless of medication state. These observations suggest that levodopamine, in contrast to DBS, may not target the hyperdirect pathway, but rather that it could target subcortical mechanisms responsible for the transduction of high beta frequencies into lower frequencies.

**Study limitations.** Our findings should be considered in light of the following limitations. Firstly, we observed phenotypic differences between the STN and GPi DBS patient groups for the UCL cohort but not for the Shanghai cohort (see "Methods" section). We accounted for phenotypic differences at UCL by including them as covariates in our statistical analyses. Importantly the fact that the results of STN versus GPi comparisons were consistent from both the Shanghai and UCL cohorts highlights that the

relative differences in observed STN and GPi activity and coupling are unlikely to have been driven by phenotypic differences.

Secondly, although we were able to validate our findings with on and off medication recordings from the STN UCL cohort, both GPi cohorts were recorded only on medication. In our experience patients undergoing GPi DBS tend to have more troublesome motor symptoms making it difficult for them to tolerate prolonged on and off medication recordings[58]. Nevertheless, in keeping with its effects on STN activity, levodopamine demonstrates suppressive effects on low but not high beta-band activity within the GPi[59]. Medication effects on cortico-pallidal synchronization at high beta frequencies are less well characterized, although our group has previously attempted to investigate this in single patient data[3]. It, therefore, remains to be determined whether akin to cortico-STN high beta coherence, cortico-GPi high beta coherence remains unaffected by medication administration.

Finally, our recordings were performed a few days after electrode implantation. It is known that the insertion of DBS electrodes can result in transient amelioration of parkinsonism and may influence LFP activity before stimulation has been started[60–62]. This may be due to the physiological effects of the lesion (a so-called 'stun effect') and also to placebo mechanisms[63]. Nevertheless, studies of LFP activity months or years after initial implantation reveal strong similarities in beta activity and in relationships between this activity and clinical state in the early postoperative and delayed postoperative periods[64–67].

## Methods

**Patients and experimental details.** Data from a total of thirty-two patients were included in this study. Twelve patients; six with bilateral implantation of STN DBS electrodes and six with bilateral implantation of GPi DBS electrodes were recruited at each of two separate university hospitals; the National Hospital for Neurology and Neurosurgery (UCL) and the Ruijin University Hospital (Shanghai JaioTong University). A further eight patients with bilateral STN electrodes from UCL were also included in the study. Patients were diagnosed with Parkinson's disease according to the Queen Square Brain Bank criteria[68].

For the UCL cohort, patients in the GPi DBS subgroup had an additional diagnosis of Parkinson's disease dementia (PDD) and were recruited into a separate trial that involved targeting both the motor GPi and the Nucleus Basalis of Meynert (NBM)[69]. Clinical characteristics of all patients are presented in Supplementary Table 1.

Phenotypic differences (Age, Disease Duration, pre-operative Levodopa equivalent dose, pre-operative UPDRS Part III motor scores ON medication, and the Mini-Mental State Examination(MMSE)) between STN and GPi DBS patients were compared using unpaired t-tests separately for the UCL and Shanghai cohorts. For the UCL cohort, the disease durations of the STN and GPi DBS patients were similar in spite of the GPi DBS subgroup being on average older. In addition, the STN subgroup had a higher levodopa equivalent dose, lower UPDRS Part III motor scores ON medication, and higher cognitive performance scores as measured by the MMSE (see Supplementary Table 1 for statistics of comparisons). In contrast for the Shanghai cohort, there were no significant group differences between STN and GPi DBS patients. In order to account for phenotypic differences for the UCL cohort, we included the four clinical features which were significantly different between the groups as covariates in all further statistical analyses.

MEG data were collected simultaneously with LFP activity recorded from DBS electrodes sited in the STN or GPi. Recordings were performed whilst patients were seated at rest either after overnight withdrawal of usual dopaminergic medication (OFF state) or approximately one hour after medication administration (ON state). Patients were examined by a movement disorders neurologist (who was present for the duration of the recordings) prior to recordings in order to ensure that they were in their usual on or off states. Patients 1–6 with STN electrodes at UCL underwent two separate recordings with randomized order: one in the OFF medication state and one in the ON medication state. Patients 7–14 with STN electrodes at UCL participated in only OFF state recordings. Patients 1–6 with GPi electrodes at UCL, patients 1–6 with STN electrodes from Shanghai, and patients 1–6 with GPi electrodes from Shanghai participated in only ON medication recordings (see Supplementary Table 1 for further details). Study procedures at UCL were approved by the Oxford B Research Ethics Committee, whilst study procedures at Shanghai were approved by the ethics committee at the Ruijin Hospital. Informed consent was sought from patients for study participation and for sharing anonymized clinical variables. Further details of the operative procedure and combined MEG-LFP recordings are found in the Supplementary Methods.

**DBS electrode localization and fiber tracking.** DBS electrodes were localized using Lead-DBS (www.lead-dbs.org). This involved linearly co-registering the post-operative MRI to the pre-operative MRI using SPM12 (Statistical Parametric Mapping; http://www.fil.ion.ucl.ac.uk/spm/software/spm12). In order to then compare electrode placement across subjects, preoperative and postoperative MRI acquisitions were non-linearly co-registered (normalization) into MNI ICBM152 NLIN 2009b stereotactic space (Montreal Neurological Institute; https://www.bic.mni.mcgill.ca/ServicesAtlases/ICBM152NLin2009) using the SPM12 segment nonlinear option in Lead-DBS. Each electrode could then be localized and visualized in the aforementioned MNI space simultaneously with masks of subcortical structures (including GPi and STN) derived from the DISTAL atlas within Lead-DBS.

For each electrode, we used an automated approach for determining which individual contacts lay within the STN or GPi. This approach relied on determining whether MNI coordinates defining individual contacts lay within a convex hull bounded by the surface of the STN or GPi (https://uk.mathworks.com/matlabcentral/fileexchange/10226-inhull). Contact pairs where at least one contact lay in the target nucleus were used for subsequent analyses; for instance, in the event of only contact 1 being inside the target nucleus, we selected contact pairs 0–1 and 1–2. For the purposes of fiber tracking, a spherical region of interest centered at the midpoint of each chosen contact pair, with a radius that just encompassed each contact pair was constructed. For STN patients 1–6 and for GPi patients 1–6 from both surgical centers this spherical volume was used as a seed region in an openly available group connectome (www.lead-dbs.org) which was derived from diffusion-weighted magnetic resonance (dMRI) images of 90 patients in the Parkinson's progression markers initiative (PPMI) database. All scanning parameters are published on the website (www.ppmi-info.org). UCL STN patients 7–14 additionally had preoperative dMRI as described in the Supplementary Methods. For both connectome and individual subject dMRI Whole brain tractography fiber sets were calculated using a generalized q-sampling imaging algorithm as implemented in DSI studio (http://dsi-studio.labsolver.org) within a white-matter mask after segmentation with SPM12. Fiber tracts were transformed into MNI space[70] for visualization. We then determined the number of fibers passing through both the aforementioned spherical seed and each cubic voxel of side 2 mm. This yielded a single number for each voxel that served as an estimate of tract density and was written to a 3D image. Prior to statistical testing images were smoothed with an 8 mm isotropic Gaussian kernel as per the analysis of MEG data.

**Analysis of oscillatory synchrony within the cortico-STN/cortico-GPi circuit.** Power spectra from STN and GPi contact pairs were computed using multitaper spectral estimation with a frequency resolution and taper smoothing frequency of 2.5 Hz[71]. Physiological power spectra may be thought of as a summation of two distinct processes: (1) an aperiodic component reflecting 1/f like characteristics which may differ across subjects and (2) periodic oscillatory components manifesting as band-limited peaks in the power spectrum. To make spectra comparable across subjects we used a spectral parameterization algorithm (Fitting Oscillations and One-Over F algorithm, https://github.com/fooof-tools/fooof) to model the aperiodic (1/f) component[72]. This was visualized in all cases for quality control and subsequently subtracted from the power spectrum in order to isolate the periodic oscillatory component of interest. To test for differences in the spectra of STN and GPi at each frequency, mean (across trials) log spectral time series were converted into 1D images, smoothed with a 2.5 Hz Gaussian kernel and subjected to a t-test within SPM. All analyses were corrected for multiple comparisons using random field theory and reported findings are significant with familywise error (FWE) correction at the cluster level ($P < 0.01$ corrected, cluster forming threshold $P < 0.001$ uncorrected). In addition to modeling subject-specific dependencies in the recordings from the two hemispheres, we included side as an additional categorical variable for each subject to account for potential differences between the recordings from the right and left sides. Covariates representing each patient's age, preoperative levodopa equivalent dose, UPDRS Part III motor score on medication, and MMSE were introduced as described above in order to account for phenotypic differences between the STN and GPi patient groups at UCL.

Brain areas coherent with STN and GPi LFPs were localized using dynamic imaging of coherent sources (DICS) beamforming[73] yielding a 3D image of coherence (see Supplementary Methods for further details). In light of previous work highlighting differences in the profile of STN-cortical connectivity in the upper (21–30 Hz) and lower (13–21 Hz) frequency bands, we restricted our DICS beamformer analysis to these two bands[4]. For each subject and each hemisphere, coherence images were generated for the two frequency bands. Half of the resulting images (all left STN/GPi images) were reflected across the median sagittal plane to allow comparison of ipsilateral and contralateral sources regardless of the original side. These images were then subjected to a 2 × 2 factorial ANOVA, with frequency (low beta versus high beta) and electrode location (STN versus GPi) as factors in SPM. Covariates were added as described above and the direction of main effects and interactions in the 2 × 2 ANOVA was tested by performing t-tests in SPM. All analyses were corrected for multiple comparisons using random field theory and reported findings are significant with familywise error (FWE) correction at the cluster level ($P < 0.01$ corrected, cluster forming threshold $P < 0.001$ uncorrected). For the STN UCL cohort, where recordings were performed both ON and OFF medication, we tested for medication and frequency interactions by performing a

separate 2 × 2 factorial ANOVA with factors frequency (low beta versus high beta) and medication state (ON versus OFF).

DICS beamformer images were also generated across the alpha (7–13 Hz) and entire beta (13–30 Hz) frequency and subjected to a separate 2 × 2 factorial ANOVA with factors frequency (alpha versus beta) and electrode location (STN versus GPi). As the aim of this analysis was to define a cortical network coupled to the STN/GPi across the entire beta frequency range—for subsequent visualization of tracts (see "Results" section)—we were primarily interested in the main effect of the band.

Next, using beamforming we performed time-series extraction from peak voxels in the SPMs of group-level main effects and interactions. Coherence was computed between the reconstructed source and the subcortical LFP using multitaper spectral estimation with a frequency resolution and taper smoothing frequency of 2.5 Hz[71]. Further details of time series analysis relating to directionality are found in the Supplementary Methods.

**Relationship of structural and functional connectivity**. For each contact pair (with at least one contact lying in the STN or GPi) in each subject, we had isotropic 2 mm 3D images of both coherence and fiber density in MNI space. Fiber density images were derived either from the PPMI connectome or from individual patient dMRI datasets for UCL STN patients 7–14. Further analysis was performed separately for each patient group (STN electrodes versus GPi electrodes) and each frequency band (low beta 13–21 Hz, and high beta 21–30 Hz) in order to establish whether tract density was predictive of coherence. As we were specifically interested in fibers representing hyperdirect connections, we minimized the chance of including fibers from the direct and indirect pathways by limiting fiber tracts to those not traversing the striatum. Accordingly, a striatal mask was created using the DISTAL atlas within Lead-DBS[74,75]. In a separate analysis, we included only fibers passing through the striatum on their passage to the STN or GPi as these may be representative of non-hyperdirect connections from the indirect and direct pathways, respectively.

For each voxel, we constructed a General Linear Model (using spm_ancova) with tract density as the independent variable and coherence as the dependent variable. Subject, side, age, medication dose, and MMSE were introduced as covariates as described above. The F-statistic was used to determine a P value, with a threshold of P < 0.01 being used to define voxels to form clusters for cluster-based permutation testing. This served to correct for multiple comparisons and we utilized a threshold of P < 0.01 to define significant clusters. The F-statistics of voxels within each significant cluster and the corresponding $R^2$ correlation coefficient were then written to a 3D image for visualization. In the further analysis we investigated fiber tracts traversing the spherical ROI associated with each contact, which started (or terminated) within cortical volumes derived from SPM analysis of the main effects of cortico-STN and cortico-GPi coherence. For each contact, we computed a single overall tract density estimate by dividing the number of fibers originating in each cortical volume by the number of voxels contained within the volume.

**Computational modeling of high and low beta band oscillatory synchrony**. A firing rate model of the cortico-basal-ganglia circuit was developed, based on models previously used to study beta oscillations[18,33]. The basic idea behind our model is to generate oscillations via the interaction of excitatory and inhibitory neural populations. Technical details are described in Supplementary Methods.

## Data availability

The MEG and LFP datasets generated within this study have not been deposited within a public repository because they contain patient-sensitive data. Anonymized datasets are available from Dr. Ashwini Oswal (ashwini.oswal@ndcn.ox.ac.uk) on reasonable request. Source data for Figs. 2, 3d–e, 4d–e, 5, 6c, and 7c are provided as a source data file. Source data are provided with this paper.

## Code availability

MATLAB code for reproducing: (1) the relationship between structural and functional connectivity, (2) computation of time delays, and (3) results of computational modeling are deposited on the GitHub repository https://github.com/AshOswal/Multimodal_Tools DOI: 10.5281/zenodo.5067980. We are grateful to Professor Karl Friston for statistical advice.

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

## Acknowledgements
A.O. is supported by an NIHR Academic Clinical Lectureship. The Wellcome Centre for Human Neuroimaging is supported by core funding from Wellcome [203147/Z/16/Z]. The work was supported by the UK MEG community Medical Research Council grant MK/K005464/1. C.C. is supported by the Natural Science Foundation of China (81571346, 82071547). A.H. was supported by the German Research Foundation (Deutsche Forschungsgemeinschaft, Emmy Noether Stipend 410169619 and 424778381—TRR 295), as well as Deutsches Zentrum für Luft-und Raumfahrt (DynaSti grant within the EU Joint Programme Neurodegenerative Disease Research, JPND). RB and PB are supported by the Medical Research Council (MC_UU_12024/5 and MC_UU_12024/1, respectively).

## Author contributions
Study design and methodology: A.O., C.C., Q.W., V.L., R.B., M.H. and P.B. Contribution of software: A.O., C.Y., W.J.N., A.H., V.L. M.E.G. and L.F.P. Data collection: A.O., J.G., D.L., S.Z., C.Z., B.S., V.L. Data analysis: A.O., and C.Y. Clinical data collection and patient characterization: A.O., C.C., H.A., D.L., Q.W., S.Z., C.Z., L.Z., T.F., P.L., B.S. Writing—original draft: A.O. and P.B. Writing—review and editing: A.O., C.C., C.Y., W.J.N., J.G., H.A., A.H., R.B., M.H., P.B., V.L. Funding acquisition: A.O., V.L., and P.B. Supervision: V.L. and P.B.

## Competing interests
The authors declare no competing interests.
