## [Peer Review File · Nature Communications]

Reviewers' comments:

Reviewer #1 (Remarks to the Author):

The manuscript „Neural signatures of pathological hyperdirect pathway activity in Parkinson’s disease” presents a work which combines different modalities and technical approaches to study connectivity in Parkinsonian patients. This includes electrophysiological data from combined LFP MEG recordings, tractography data from a database of MRI scans, and mathematical modelling. The authors provide evidence for a correlation between functional coupling as revealed by electrophysiological data with anatomical connectivity as shown by tractography. They substantiate this finding by a sophisticated and plausible computational model. They interpret their findings as a signature of the hyperdirect pathway being active at beta frequency range and argue that this observation suggests a pathophysiological role of the hyperdirect pathway in oscillatory network dysfunction in Parkinson’s disease.

The present work is a useful, interesting, and timely approach requiring broad expertise and significant effort. However, this broad approach also might make it difficult for readers to comprehend the content. In particular, this will be the case, if readers do not have deep and broad knowledge in the field. The complexity of the manuscript at hand is also reflected by the extensive supplementary material provided by the authors. Therefore, the manuscript is not an easy read, although it is clearly written. On the other hand, it cannot be shortened without compromising reproducibility.

The work is a logical progression and integration of earlier work by the authors. The integration in the relevant literature is fair and sufficient and reflects an excellent knowledge of the state-of-the-art. The study combines and extends previous results by the authors and others in the field elegantly. This combination of technical approaches is the major merit and new achievement of the work at hand, while the novelty of the actual findings and claim is limited. However, the integration of different data and modelling makes the work interesting and relevant for people working in the field and it will potentially influence thinking in the field.

Currently, there are weaknesses in the work and the most significant one is that the number of patients under study is very small. While it needs to be acknowledged that such recordings are hard to realise and that this data is of great value, such a low number of patients is problematic and has relevant implications and limitations for the statistical approaches being used. I’m not sure that the approach used here is the most adequate one. Therefore, additional consultation by an expert in statistics would be useful. This is particularly the case for the comparison of the two groups and the use of an ANOVA with such a small sample. I’m not sure if a mixed linear model would not be more adequate. In addition, correlations with such a small sample need to be interpreted with caution. Moreover, it has to be considered that activity from two STNs/GPis of a single patient cannot be considered as independent samples per se. The same is true for the contacts of one electrode and within one patient. Furthermore, it would be useful to report effect sizes in addition to p-values. Taken together, inclusion of more patient data in the present work would constitute a significant and effective strengthening of the present study.

Further limitations of the work are clearly addressed by the authors themselves in the discussion. Here, an important limitation is that the two groups under study (STN patients vs. GPI patients) differ in several aspects (e.g. medication doses, UPDRS, cognitive assessment). It is not clear how much these differences affect the present findings.

Moreover, the information on clinical characteristics is sparse. For example, information of medication during recordings is missing. Also, no information about handedness, symptom dominant side, and sex is provided.

Please note, that my expertise on computational modelling is rather limited and I can only review this part of the manuscript to a quite basic level.

All in all, this manuscript is a very interesting and relevant approach to an important research question, but at the same time, it comprises some weaknesses needing further attention by the authors of which the small sample size is the most relevant one currently.

Reviewer #2 (Remarks to the Author):

Oswal et al. utilize an impressive number of approaches (DBS LFP, MEG, tractography analysis and computational modelling) in an effort to support an interesting hypothesis that pathological

synchronization in PD arises from the cortico-STN hyperdirect pathway. The idea that hyperdirect connectivity in high beta band drives additional low beta pathological synchrony is novel and interesting, though this finding is supported only by modelling. The paper is well written, though details of the statistical underlying main claims can be somewhat difficult to follow. This paper would definitely be of interest to people in the field of PD and DBS, and drive additional follow-up studies and hypotheses building upon the work presented here. There is some question whether the data provided are sufficient to make the claims of the paper as strong as they are. The findings of the study could be greatly enhanced by inclusion of off-medication recordings, if they were collected in these patient cohorts. There are a number of concerns that could be addressed to strengthen the paper:

Major concerns:

Line 89: "Cortico-STN anatomical connectivity derived from individual patient electrode localisations and open source tractography connectomes ... " The study would be greatly improved if individual high resolution MRI imaging to develop patient-specific tractography, though such imaging may not have been acquired. Patient specific fibre tractography would seem necessary in order to "establish the relationship between hyperdirect pathway fibre density and cortico-STN coupling at high beta frequencies." The results of this study depend in part on the assumption that the fibre tractography based on the lead-DBS connectome accurately reflects the target structure and tractography of all patients, though significant subject-subject variability is known (e.g. Patriat 2018 <https://doi.org/10.1016/j.neuroimage.2018.05.048>)

Methods: "Recordings were performed at rest whilst patients were on their usual medication." Were some recordings not made after withdrawal of medication? What would high beta activity and cortical – basal ganglia synchrony look like in the "parkinsonian" off-medication state? Including a comparison between the off and on medication state would be highly informative. If high beta coherence between GPi and cortex emerged in the off-medication state, how would that change (if at all) the author's interpretation of the main findings of this study? See Figure 8, Williams et al. 2002, <https://doi.org/10.1093/brain/awf156>, where GPi-Cx high beta coherence is present off medication but minimal on medication. The fact that recordings were made on medication should be mentioned in the main text, in the results, as this is important for interpretation, not just stated in the methods.

Fig. 2. There is large variability in the LFP power spectra in STN and GPi, which is not surprising as many studies observe large subject-subject variability in these DBS LFP measures, possibly due to lead location differences, differences in patient phenotype, or other factors. The authors appropriately display PSD traces from all subjects, however they then present mean PSD and SEM (? - shaded region is not defined in the figure legend). The authors should make a compelling rationale as to why using mean across the population is appropriate to describe the data. Is it appropriate to use a t-test given the characteristics of these PSD distributions? They do not appear to be normally distributed. Combined with apparent large variability in lead location in Fig. 1, it would be useful to associate individual lead locations in Figure 1 and Supplementary Figures 1 & 2 with corresponding spectral profiles for each patient. In addition, similar to the PSD plots, including coherence plots from each subject is warranted to illustrate variability across the patient population (Fig 3).

The authors presume that the hyperdirect pathway is exaggerated in PD. They should also acknowledge the literature that is contrary to this idea (e.g. Chu et al. 2017 <https://doi.org/10.1016/j.neuron.2017.08.038>, Matthai et al. 2015 <https://doi.org/10.1093/brain/awv018>). The authors should probably temper the claim made in their title "Neural signatures of pathological hyperdirect pathway activity in Parkinson's disease". It is not apparent that the authors present data that shows that the high beta cortico-STN coherence observed is in fact pathological, particularly given that recordings were made on medication.

Additional comments:

Lines 72-79: While the statement line 74 regarding lower beta suppression and its relationship to

clinical improvement is valid and important, it is unclear how those findings “lead to the hypothesis that cortical coupling with the STN at high beta frequencies may reflect hyperdirect pathway activity”. Some rewording / reorganizing of these sentences may be appropriate.

Lines 79-81: “If high beta frequencies are reflective of the hyperdirect pathway, synchrony within this frequency range should be relatively restricted to the STN and its interactions with cortex.” High beta band activity has been observed outside of the STN, and high beta coherence has been observed between cortex and GPi (Williams et al. 2002, <https://doi.org/10.1093/brain/awf156>, see also Talakoub et al. 2016, <https://doi-org.ezp3.lib.umn.edu/10.1038/srep34930>). Given these and other studies, it would seem that synchrony in this frequency range is not restricted to the STN and cortex. At least authors should acknowledge some of the literature that already addresses and may not necessarily support this hypothesis (that high beta synchrony is mostly restricted to STN and cortex)

Lines 83-85: “...STN beta waveform features suggestive of synchronisation 29,30 to a cortical source...” While cited studies indicate waveform shape is suggestive of synchronisation, can authors assert that it the source is necessarily cortical? Could it not reflect synchronous inputs from other connected sites (e.g. GPe, CM/PF).

Line 118: MNI has not been previously defined

Figure 3 D-F; it is not readily apparent that the black arrows correspond to distinct peaks in the coherence plots. There is a broad flat peak in cortico-STN coherence; why is that particular point with black arrow highlighted? The low frequency arrow in D does not appear to be a distinct peak. Why does panel F lack an arrow?

Rest recordings had a duration of 3 minutes. How stationary were the recordings across time (both PSD and coherence measures)? Is it feasible to divide the recording duration into smaller segments and average (therefore possibly less susceptible to brief epochs that might be outliers)?

The clinical characteristics between the STN and GPi cohorts appear quite different (e.g. UPDRS-III scores). Could differences in patient phenotype underlie the differences observed in high beta, rather than differences between STN and GPi connectivity? The authors discussion of this being a study limitation is appreciated, however the cohorts are receiving very different amounts of levodopa; again, if any data were collected off medication in these patient cohorts, analysis with those data would be very beneficial. Although on-medication data collection is mentioned as a limitation, it perhaps should not be dismissed saying high beta is unlikely to be impacted. Though referenced studies may show minimal impact in the STN, the literature on GPi DBS LFPs off and on medication is more sparse, and as already mentioned has been shown to impact high beta GPi coherence measures in previous studies.

Line 166: “We characterised the directionality of cortico-subcortical coupling at beta frequencies in order to determine whether these signals originate from the cortex or subcortical structures.” Stronger Cortex->STN Granger causality measures do not necessarily imply that is due to the hyperdirect pathway, does it? Could it not be striatal in origin via the indirect pathway?

Do the time delays presented (20ms) support the argument that the hyperdirect pathway is driving this connectivity? I would point out the following statement from Williams et al. 2002: “Where cortex drives STN this could be achieved either directly through the large cortico-subthalamic projection, or indirectly via the putamen/globus pallidus externa (Parent and Hazrati, 1995). A delay of ~20 ms would seem to favour the latter”

Figure 5: To provide the reader with greater intuition about the finding “tract density predicts high beta coherence,” showing data across patients (e.g. scatterplot of fiber density vs high beta coherence across patients) might be helpful.

Lines 334-337: The hyperdirect cortico-STN connections are not the only difference in connectivity between STN and GPi; it is a leap to assume that “by comparing LFPs from patients with STN electrodes to those from patients with GPi electrodes, we were able to isolate electrophysiological

markers of hyperdirect pathway activity." This seems too strong a statement.

Although there are many figures and the current manuscript is extensive, the results of this study may be stronger if it included data during a motor task (though perhaps this was not collected), which might engage the hyperdirect pathway more distinctly than at rest.

Supplementary Methods

Please define/describe MNI space

Line 94: should read duration \geq 400ms ?

What was the magnet strength on the MRI imaging?

Reviewer #3 (Remarks to the Author):

The main hypothesis of this paper that cortical coupling with the STN at high beta frequencies reflects hyperdirect pathway activity, that synchrony within this frequency range should be restricted to the STN and its interactions with cortex, and the authors predict an overlap between cortico-STN anatomical connectivity of the hyperdirect pathway and the profile of cortico-STN functional connectivity at high beta frequencies. This is an interesting thesis that the authors have developed from previous work and may be very important in the field.

Despite the statistical methods to correct for the multiple variables that could contribute to the results it is difficult to judge the interpretation of the results due to several confounding variables that will affect the LFP recorded: the participants were all on medication, the lead and electrode pair location varied in both the STN and GPi, the cohort was small and the GPi group had been chosen for the NBM target and were cognitively impaired. If any or many of these factors altered the LFP recorded it would be subject specific and hard to correct for in my opinion.

Major concerns

-Small heterogeneous cohort, 6 patients with GPi DBS and 6 patients with STN DBS are used in the study. There is no mention of N in the abstract, nor the number data points used for each result. There is a difference in cognitive scores, as the cohort used for GPi appear to be the cohort used for the NBM target for PD dementia. Can the authors confirm that the relative lack of power in the high beta range in the GPi cohort was not influenced by cognitive impairment or the effect of medication?

-Another concern is that the study was done on medication and the LEDD varied among people and between targets: LEDD= 1149mg for the STN group and 647mg for the GPi group. The report that is was not significant attests to the variation among subjects. This will result in attenuation of beta power that would be heterogeneous but not related to whether it was GPi or STN. The time course of the recording after a dose of medication will also be different and did not seem to be standardized. This will affect the LFP and makes the results hard to interpret, and it would not be corrected just by accounting for LEDD as a covariate as the degree of attenuation of beta power is not linear and is very subject and timing specific. Please clarify whether the UPDRS III and LEDD data were the pre-operative values or at the time of recording and how many days post lead implantation the studies were done. Was this different regarding the potential effect of the lesion effect on LFP power?

- It is unclear for many of the results what data are being included. Based on the methods (e.g., ~lines 506-511 on pg 22 and ~ line 32 on pg 2 of the supplemental data), it appears that multiple channel pairs from the same hemisphere are being used for some participants. If one STN had multiple electrode pairs used this could bias the finding of high beta power, despite correction, especially given the small sample of 6 patients per group for any conclusions that are being drawn. For instance the high beta power in the STN spectra in Fig 2B appear to be driven by a few spectra and if these are all from the same nucleus this would bias the results. It would also be clearer to

the reader if individual data points were superimposed on the bar graphs and that it was clearly identified which and how many channel pairs were being used per nucleus.

-Was there any difference in high beta power between the STN and GPI data if the authors did not subtract the $1/f$ curve and can they provide evidence that this is a robust method that would not vary based on all of the variables above?

- Was the power in Fig 2 normalized so it could be compared across STNs/GPis?

- lead location variability

In addition to the small number of nuclei/subjects the varying locations of the DBS leads in the STN and the GPI also introduce variability in the LFP. The GPI placement was targeting the NBM and not sensorimotor GPI and this might alter the LFP spectra. The HDP and other afferent inputs are likely conserved anatomically and so recordings from a variety of locations may contribute to variability in results.

- Latency of the putative HDP

This perhaps may have been part of a puzzling result regarding monosynaptic latencies. Does it seem concerning to the authors that the observed time-delay between the STN and cortical sources is so long (>15 ms)? This is not on the time-scale that would be expected for a monosynaptic connection such as the hyperdirect pathway. Cortical evoked potentials following stimulation of the STN were recently shown to take only ~ 2 ms (see Chen et al., 2020 "Prefrontal-Subthalamic Hyperdirect Pathway Modulates Movement Inhibition in Humans"). With the functional connectivity analyses, there is no way to know if there is a 3rd (or more) region mediating the connection and the observed time-delay seems to highlight that concern.

- Non-linear waveform interpretation and place in ms

It is slightly unclear what the purpose of the investigation of the nonlinear waveform features of the two regions is within the current experimental framework. These results are mentioned in the abstract and get a substantial section in the discussion, but then are actually only placed in the supplementary materials. Although there is certainly a growing interest in investigating the sharpness of beta oscillations (especially in PD), the authors argue that the observed difference in sharpness is reflective of a difference in synchronization of input. The actual direct evidence for this is still extremely limited and it is a large jump to take the observed differences in sharpness observed in the STN vs. GPI to say this is evidence for a "dominance of a very direct pathway from cortex to STN".

- Figure 3 – can the authors clarify whether coherence would be affected by power? If there was greater high beta power in the STN than the GPI then would that influence the greater coherence between the cortex and STN compared to the cortex and GPI in high beta?

- line 286, could the authors list a range of the cortical spike activity (I assume they mean 650 spikes/s or more after the bifurcation?)

-It is apparent how changing the feedback loop delay changes high and low beta, but how do the model results directly relate to coherence?

In what follows, the reviewers' questions are in bold font, our responses are in plain font and references to appropriate sections of our revised manuscript are in quotation marks with changed sections in italic font.

Reviewer 1

The manuscript ‘Neural signatures of pathological hyperdirect pathway activity in Parkinson’s disease’ presents a work which combines different modalities and technical approaches to study connectivity in Parkinsonian patients. This includes electrophysiological data from combined LFP MEG recordings, tractography data from a database of MRI scans, and mathematical modelling. The authors provide evidence for a correlation between functional coupling as revealed by electrophysiological data with anatomical connectivity as shown by tractography. They substantiate this finding by a sophisticated and plausible computational model. They interpret their findings as a signature of the hyperdirect pathway being active at beta frequency range and argue that this observation suggests a pathophysiological role of the hyperdirect pathway in oscillatory network dysfunction in Parkinson’s disease.

The present work is a useful, interesting, and timely approach requiring broad expertise and significant effort. However, this broad approach also might make it difficult for readers to comprehend the content. In particular, this will be the case, if readers do not have deep and broad knowledge in the field. The complexity of the manuscript at hand is also reflected by the extensive supplementary material provided by the authors. Therefore, the manuscript is not an easy read, although it is clearly written. On the other hand, it cannot be shortened without compromising reproducibility.

The work is a logical progression and integration of earlier work by the authors. The integration in the relevant literature is fair and sufficient and reflects an excellent knowledge of the state-of-the-art. The study combines and extends previous results by the authors and others in the field elegantly. This combination of technical approaches is the major merit and new achievement of the work at hand, while the novelty of the actual findings and claim is limited. However, the integration of different data and modelling makes the work interesting and relevant for people working in the field and it will potentially influence thinking in the field.

We thank the reviewer for their supportive comments and appreciation of the complexity of the datasets.

1) Currently, there are weaknesses in the work and the most significant one is that the number of patients under study is very small. While it needs to be acknowledged that such recordings are hard to realise and that this data is of great value, such a low number of patients is problematic and has relevant implications and limitations for the statistical approaches being used. I’m not sure that the approach used here is the most adequate one. Therefore, additional consultation by an expert in statistics would be useful. This is particularly the case for the comparison of the two groups and the use of an ANOVA with such a small sample. I’m not sure if a mixed linear model would not be more adequate. In addition, correlations with such a small sample need to be

interpreted with caution. Moreover, it has to be considered that activity from two STNs/GPis of a single patient cannot be considered as independent samples per se. The same is true for the contacts of one electrode and within one patient.

We thank the reviewer for these very important points. In response to the concerns about sample size, in our revised manuscript we have included data from a further 12 patients (6 with GPi electrodes and 6 with STN electrodes) that were collected independently by our collaborators at the Ruijin University Hospital in Shanghai. The recordings from Shanghai were collected ON medication. Additionally for the STN UCL patients 1-6 we have included both ON and OFF medication recordings; four patients that were previously in this group but had both ON and OFF medication recordings were retained and two patients with only ON medication recordings were replaced by two patients with ON and OFF recordings.

Importantly, we have been able to replicate many of our own findings from UCL in the Shanghai cohort which serves to strengthen conclusions about the experimental findings. We choose to analyse the Shanghai and UCL datasets separately, rather than merging them, since replication of the findings in two different samples is a more stringent test than replication by increasing sample size alone – as the latter may bias towards smaller effect sizes.

Additionally we have included data from a further 8 STN DBS patients who were recruited at UCL and recorded off medication. These subjects underwent both MEG-LFP recordings and preoperative diffusion weighted MRI meaning that we were able to validate our observed relationships between MEG derived measures and tractography connectomes on individual subject tractography data. Our total sample size has therefore now increased to 32 patients.

The statistical inference approaches used in the present manuscript are well established in the neuroimaging statistics literature¹⁻⁴. We have utilised the Statistical Parametric Mapping (SPM) approach and permutation tests in order to correct for multiple comparisons⁴⁻⁶. With regard to reviewer's specific point about the use of ANOVA, it is important to point out that this test makes no assumptions about sample sizes⁷. Furthermore, smaller sample sizes necessitate larger effect sizes for differences to reach a given significance threshold. The corollary is that large sample sizes can bias towards trivial effect sizes which is both a problem in many studies and a fallacy of classical statistical tests⁷.

With regard to the reviewer's point about mixed effects analyses or accounting for dependencies in recordings from the same subject and in recordings from each hemisphere of each subject, we apologise for our imprecise description. We actually used covariates in our general linear model design matrix, in order to model subject and side dependencies. This meant that recordings from the same electrode and from different hemispheres of a patient were not treated as independent. This is a standard approach in second level analyses of M/EEG data, where a summary statistic (in this case the mean) from each subject undergoes standard general linear model (GLM) analysis^{1,8}.

In order to clarify these issues the following statements have been added to the methods, under the heading, '*Analysis of oscillatory synchrony within the cortico-STN/cortico-GPi circuit*':

'To test for differences in the spectra of STN and GPi at each frequency, mean (across trials) log spectral timeseries were converted into 1D images, smoothed with a 2.5 Hz gaussian kernel and subjected to a t-test within SPM. All analyses were corrected for multiple

comparisons using random field theory and reported findings are significant with familywise error (FWE) correction at the cluster level ($P < 0.01$ corrected, cluster forming threshold $P < 0.001$ uncorrected). In addition to modelling subject-specific dependencies in the recordings from the two hemispheres, we included side as an additional categorical variable for each subject to account for potential differences between the recordings from the right and left sides. Covariates representing each patient's preoperative levodopa equivalent dose, UPDRS Part III motor score on medication and MMSE were introduced as described above in order to account for phenotypic differences between the STN and GPi patient groups at both surgical centres.'

Finally, in line with the reviewer's advice we now also sought validation of our statistical procedure from a colleague, Professor Karl Friston, who is an international expert in Brain imaging statistics and a developer of the SPM software package. We have acknowledged his important contribution in the revised Acknowledgements section of our manuscript.

2) Furthermore, it would be useful to report effect sizes in addition to p-values. Taken together, inclusion of more patient data in the present work would constitute a significant and effective strengthening of the present study.

We thank the reviewer for these important points. We have now ensured that in all figures we show mean profiles and their standard errors which serve as an indicator of effect sizes (Please see revised Figures 2, 3, 4, 5, 6, 7). Importantly in Figures 3 and 4 which show whole brain T-statistics we have performed beamformer source extractions for the locations of the peak T-statistics so that readers are able to visualise the magnitude of the coherence effects. This is described in the revised Results section as follows (see third paragraph under heading, '*Cortical-subcortical coherence at high beta frequencies occurs preferentially within the cortico-STN network*')

*'To further visualise these effects, and provide indication of effect sizes, source extracted coherence spectra are shown in **Figures 3D-E**. **Figures 3D** and **3E** show coherence profiles of the STN and GPi for the peak locations within the mesial anterior (SMA) and mesial posterior (mesial primary motor cortex, M1) clusters for which there was a simple main effect of band for the STN (visualised in **Figure 3B**).'*

Additionally, to display effect sizes for the tractography-coherence correlations in Figures 6 & 7 we have added R^2 correlation maps to the previously displayed volumetric T statistic images. These display the correlation coefficient at each voxel within each significant cluster for which there was a correlation between high beta band coherence and tract density. This is now also described in the Methods section under the heading '*Relationship of structural and functional connectivity*' as follows:

'The F -statistics of voxels within each significant cluster and the corresponding R^2 correlation coefficient were then written to a 3D image for visualisation.'

The Results section has also been updated. Please see the following sentences under the heading, '*Functional connectivity is predicted by anatomical connectivity within the cortico-STN hyperdirect pathway*':

*' R^2 correlation coefficient maps for voxels within each significant cluster are displayed in **Figure 6B**.'*

And:

*'Electrode localisations for this cohort are displayed in **Figure 7A**, whilst **Figure 7B** displays the corresponding F statistics and R^2 correlation maps.'*

In response to the reviewer's final point about sample size, As outlined in response 1, our sample size has now significantly increased to a total of 32 patients.

3) Further limitations of the work are clearly addressed by the authors themselves in the discussion. Here, an important limitation is that the two groups under study (STN patients vs. GPi patients) differ in several aspects (e.g. medication doses, UPDRS, cognitive assessment). It is not clear how much these differences affect the present findings.

We are grateful to the reviewer for this very important point. There were indeed phenotypic differences between the STN and GPi groups for the UCL cohort. In the newly included Shanghai cohort however the groups are much more closely matched (see updated Supplementary Table 1). The finding that results from the UCL cohort closely match those from the Shanghai cohort indicates that phenotypic differences are unlikely to be responsible for the group differences observed.

Additionally, for the UCL cohort where phenotypic differences were observed we accounted for them in statistical analyses by including them as covariates. This is statistically equivalent to 'regressing out' the effects of variables for which there were phenotypic differences before testing for group differences. We have explained this in the updated Methods section as follows (please see third paragraph of Methods section which reads as follows):

'Phenotypic differences (Age, Disease Duration, pre-operative Levodopa equivalent dose, pre-operative UPDRS Part III motor scores ON medication and the Mini-Mental State Examination(MMSE)) between STN and GPi DBS patients were compared using unpaired t -tests separately for the UCL and Shanghai cohorts. For the UCL cohort, the disease durations of the STN and GPi DBS patients were similar in spite of the GPi DBS subgroup being on average older. Additionally the STN subgroup had a higher levodopa equivalent dose, lower UPDRS Part III motor scores ON medication and higher cognitive performance scores as measured by the MMSE (see Supplementary Table 1. For statistics of comparisons). In contrast for the Shanghai cohort, there were no significant group differences between STN and GPi DBS patients. In order to account for phenotypic differences for the UCL cohort, we included the four clinical features which were significantly different between the groups as covariates in all further statistical analyses.'

Additionally we have updated the discussion of group differences in the Discussion as follows (see section titled 'Study Limitations'):

*'Our findings should be considered in light of the following limitations. Firstly, we observed phenotypic differences between the STN and GPi DBS patient groups for the UCL cohort but not for the Shanghai cohort (see **Methods**). We accounted for phenotypic differences at UCL by including them as covariates in our statistical analyses. Importantly the fact that the results of STN vs. GPi comparisons were consistent from both the Shanghai and UCL cohorts*

highlights that the relative differences in observed STN and GPi activity and coupling are unlikely to have been driven by phenotypic differences.'

4) Moreover, the information on clinical characteristics is sparse. For example, information of medication during recordings is missing. Also, no information about handedness, symptom dominant side, and sex is provided.

We thank the reviewer for this suggestion. In response we have now updated Supplementary Table 1 with the clinical information requested.

We are grateful to the reviewer for their thoughtful comments and suggestions.

Reviewer 2

Oswal et al. utilize an impressive number of approaches (DBS LFP, MEG, tractography analysis and computational modelling) in an effort to support an interesting hypothesis that pathological synchronization in PD arises from the cortico-STN hyperdirect pathway. The idea that hyperdirect connectivity in high beta band drives additional low beta pathological synchrony is novel and interesting, though this finding is supported only by modelling. The paper is well written, though details of the statistical underlying main claims can be somewhat difficult to follow. This paper would definitely be of interest to people in the field of PD and DBS, and drive additional follow-up studies and hypotheses building upon the work presented here.

We thank the reviewer for these supportive comments.

There is some question whether the data provided are sufficient to make the claims of the paper as strong as they are. The findings of the study could be greatly enhanced by inclusion of off-medication recordings, if they were collected in these patient cohorts. There are a number of concerns that could be addressed to strengthen the paper:

Major concerns:

1) Line 89: “Cortico-STN anatomical connectivity derived from individual patient electrode localisations and open source tractography connectomes ... “ The study would be greatly improved if individual high resolution MRI imaging to develop patient-specific tractography, though such imaging may not have been acquired. Patient specific fibre tractography would seem necessary in order to “establish the relationship between hyperdirect pathway fibre density and cortico-STN coupling at high beta frequencies.” The results of this study depend in part on the assumption that the fibre tractography based on the lead-DBS connectome accurately reflects the target structure and tractography of all patients, though significant subject-subject variability is known (e.g. Patriat 2018 <https://doi.org/10.1016/j.neuroimage.2018.05.048>)

We thank the reviewer for raising this very important point. Our core reason for using connectome data was that recent work (published after submission of this paper) including

co-authors of the present paper has demonstrated that the PPMI connectome gives very similar profiles of cortico-STN structural connectivity compared to individual subject diffusion MRI (dMRI) datasets⁹. It turns out that both types of connectome can be used to accurately predict responses to stimulation⁹. Furthermore, preoperative dMRI data are not routinely acquired in DBS patients and cannot be easily acquired postoperatively without substantial constraints, meaning that connectome data is often leveraged. There have been a number of studies highlighting that connectome data can be successfully leveraged to predict optimal connectivity profiles, clinical outcomes and behaviour⁹⁻¹³.

Although connectome data may potentially have greater signal to noise⁹ we fully appreciate the reviewer's point that they may not capture the full extent of inter-subject variability. For this reason we have added data from a further 8 patients with STN electrodes from UCL where both individual dMRI and post-operative MEG-LFP recordings were performed. The MEG-LFP data were recorded in the off medication state but we also separately show (see response to point 1) that there are no significant differences in cortico-STN coherence on and off medication. Crucially, using this dataset with individual subject connectomes we are able to replicate our core finding that hyperdirect pathway fibre densities correlate strongly with SMA-STN high beta band coherence.

Details of the individual dMRI processing have been added to the Supplementary Methods under the heading, '*Preoperative diffusion MRI acquisition and pre-processing*'. The Results section has also been updated with the new analyses, see the 4th paragraph under the heading '*Functional connectivity is predicted by anatomical connectivity within the cortico-STN hyperdirect pathway*':

*'Finally for the separate cohort of UCL patients with both individual structural connectomes and combined MEG-LFP recordings (UCL STN patients 7-14; see **Supplementary Table 1**), we observed an almost identical relationship between structural and functional connectivity within the hyperdirect pathway. Electrode localisations for this cohort are displayed in **Figure 7A**, whilst **Figure 7B** displays the corresponding *F* statistics and *R*² correlation maps....'*

2) Methods: "Recordings were performed at rest whilst patients were on their usual medication." Were some recordings not made after withdrawal of medication? What would high beta activity and cortical – basal ganglia synchrony look like in the "parkinsonian" off-medication state? Including a comparison between the off and on medication state would be highly informative. If high beta coherence between GPi and cortex emerged in the off-medication state, how would that change (if at all) the author's interpretation of the main findings of this study? See Figure 8, Williams et al. 2002, <https://doi.org/10.1093/brain/awf156>, where GPi-Cx high beta coherence is present off medication but minimal on medication. The fact that recordings were made on medication should be mentioned in the main text, in the results, as this is important for interpretation, not just stated in the methods.

We are grateful for this suggestion. In response to the reviewer's point we have been able to include on and off medication data for patients undergoing STN DBS at UCL (STN UCL patients 1-6 in updated Supplementary Table 1). We have also replicated many of our findings by including data from a further 12 patients (6 with GPi electrodes and 6 with STN electrodes) that were collected on medication independently by our collaborators at the Ruijin

University Hospital in Shanghai (see also response 1 to Reviewer 1). We have explained this in our revised methods section as follows (see third paragraph of Methods which includes):

*'Recordings were performed whilst patients were seated at rest either after overnight withdrawal of usual dopaminergic medication (OFF state) or approximately one hour after medication administration (ON state). Patients were examined by a movement disorders neurologist (who was present for the duration of the recordings) prior to recordings in order to ensure that they were in their usual on or off states. Patients 1-6 with STN electrodes at UCL underwent two separate recordings with randomised order: one in the OFF medication state and one in the ON medication state. Patients 7-14 with STN electrodes at UCL participated in only OFF state recordings. Patients 1-6 with GPi electrodes at UCL, patients 1-6 with STN electrodes from Shanghai and patients 1-6 with GPi electrodes from Shanghai participated in only ON medication recordings (see **Supplementary Table 1** for further details).'*

Additionally we have updated the Results section to include details of the on vs. off medication comparisons for the STN UCL cohort.

Please see the following under the heading, *'Differences in local synchrony between the STN and GPi and medication effects on the STN at beta frequencies'*:

*'For the Shanghai STN and GPi cohort and for the UCL GPi cohort recordings were only performed ON medication. In contrast, for the UCL STN cohort in which patients underwent both ON and OFF medication recordings, we observed a region within low beta frequencies (**yellow line in Figure 2B**; peak $t = 4.24$, FWE $p = 3 \times 10^{-3}$) where low beta power was increased in the OFF state compared to the ON state. Importantly there were no medication effects on high beta power within the STN.'*

Additionally for the STN UCL cohort there were no effects of medication on cortico-STN coherence, directionality or on the relationship between tract density and high beta band coherence. Please see the following statements in our revised manuscript:

Firstly, in the Results section under the heading, *'Cortical-subcortical coherence at high beta frequencies occurs preferentially within the cortico-STN network'*:

*'Finally for the STN UCL cohort comparison of cortico-STN coherence profiles ON and OFF medication revealed only a main effect of band, such that mesial motor areas including SMA were preferentially coupled to the STN at high rather than at low beta frequencies (**Figure 3C**; peak $t = 5.24$, FWE $p = 4 \times 10^{-3}$, at MNI co-ordinates 16 12 70). We observed no significant main effect or interaction of medication state.'*

Secondly, please refer to the following in the results section, under the heading, *'Functional connectivity is predicted by anatomical connectivity within the cortico-STN hyperdirect pathway'*:

*'The results of cluster based permutation testing are shown in **Figures 6A**, and reveal a lateralised cluster encompassing the SMA (blue contour), where tract density was predictive of high beta band coherence for the UCL dataset both ON and OFF medication and for the Shanghai dataset.'*

Finally the analysis of medication effects is now summarised in the revised Discussion in the section titled, '*Medication effects on high and low beta band power and coherence*':

For the UCL STN cohort, in keeping with previous reports, we observed no medication effects on coherence or on power within the high beta frequency range^{2,29,53,54}. Levodopa administration did however lead to a significant reduction in low beta band power within the STN as previously described^{22,27,29}. Importantly the relationship between hyperdirect pathway tract density and high beta band cortico-STN coherence was maintained regardless of medication state. These observations suggest that levodopamine, in contrast to DBS, may not target the hyperdirect pathway, but rather that it could target subcortical mechanisms responsible for the transduction of high beta frequencies into lower frequencies.

Unfortunately we have not been able to perform simultaneous MEG and LFP recordings both on and off medication in patients undergoing GPi DBS. In our experience this patient group tends to be more difficult to study than patients undergoing STN DBS as their motor symptoms can be more severe. We have included this as a limitation in the section titled '*Study Limitations*' within the Discussion:

'Secondly, although we were able to validate our findings with on and off medication recordings from the STN UCL cohort, both GPi cohorts were recorded only on medication. In our experience patients undergoing GPi DBS tend to have more troublesome motor symptoms making it difficult for them to tolerate prolonged on and off medication recordings⁵⁸.

We also thank the reviewer for pointing us to the paper by Williams et al¹⁴, which is work performed by members of our group a number of years ago. Importantly Figure 8 from Williams et al. shows coherence between a single EEG bipolar channel (very limited cortical sampling) and a single GPi bipolar channel in just one subject (N=1). It would therefore not be possible to draw any robust inferences about medication effects on GPi coherence and power from this report alone. To our knowledge there are no more recent reports investigating medication effects on cortico-GPi coherence with larger sample sizes. We did however identify one report by Lofredi¹⁵ and colleagues showing that dopaminergic medication has similar effects on activity within the GPi and the STN, in that it selectively suppresses low but not high beta band power.

We have now included this discussion (with reference to the paper by Williams et al.) in the section titled '*Study Limitations*' within the Discussion:

Nevertheless, in keeping with its effects on STN activity, levodopamine demonstrates suppressive effects on low but not high beta band activity within the GPi⁵⁹. Medication effects on cortico-pallidal synchronisation at high beta frequencies are less well characterised, although our group has previously attempted to investigate this in single patient data³. It therefore remains to be determined whether akin to cortico-STN high beta coherence, cortico-GPi high beta coherence remains unaffected by medication administration.

3) Fig. 2. There is large variability in the LFP power spectra in STN and GPi, which is not surprising as many studies observe large subject-subject variability in these DBS LFP measures, possibly due to lead location differences, differences in patient phenotype, or other factors. The authors appropriately display PSD traces from all

subjects, however they then present mean PSD and SEM (? - shaded region is not defined in the figure legend). The authors should make a compelling rationale as to why using mean across the population is appropriate to describe the data. Is it appropriate to use a t-test given the characteristics of these PSD distributions? They do not appear to be normally distributed.

We thank the reviewer for raising these interesting points. In the present report we have performed statistical testing on log transformed power spectra (please see revised Figures 2 and 3). Previous simulations of electrophysiological data have demonstrated that parametric tests on both log transformed and untransformed power spectra (which adopt a chi-squared distribution) are valid and robust in terms of controlling false positive rates (please see Figure 5 within the following paper <https://www.ncbi.nlm.nih.gov/pmc/articles/PMC6871741/>)¹⁶.

The following revised statement which describes log transformation in the Methods section, under the heading, '*Analysis of oscillatory synchrony within the cortico-STN/cortico-GPi circuit*':

'To test for differences in the spectra of STN and GPi at each frequency, mean (across trials) log spectral timeseries were converted into 1D images, smoothed with a 2.5 Hz gaussian kernel and subjected to a t-test within SPM.'

It is also worthwhile pointing out that it is not the data that are assumed to be Gaussian with parametric tests (such as the t-test), but the distribution of the random errors⁷. These are generally considered to be Gaussian, (by Central Limit Theorem) because of smoothing applied to the data and because summary statistics at the between subject level are linear mixtures of data at the within subject level. A final and noteworthy point is that parametric tests have been shown in simulation to be very robust to violations of gaussianity assumptions^{16,17}.

Further technical details regarding the validity of assumptions of parametric testing and their specific use in the SPM software package (also with applications to electrophysiological datasets such as our own) can be found in separate manuscripts^{1,3,7}.

4) Combined with apparent large variability in lead location in Fig. 1, it would be useful to associate individual lead locations in Figure 1 and Supplementary Figures 1 & 2 with corresponding spectral profiles for each patient. In addition, similar to the PSD plots, including coherence plots from each subject is warranted to illustrate variability across the patient population (Fig 3).

We thank the reviewer for these suggestions. In our revised coherence plots in Figures 3 and 4 we have included data from all patients (and bipolar contacts) as advised.

Regarding associating individual contacts in each subject with spectral profiles, we did not find a simple way to do this without using multiple colours – a different colour for each of the 12 subjects - which resulted in a rather messy and uninterpretable figure. We would also like to point out that the majority of contacts were well sited within the motor STN or the motor GPi. Please see Figure 1, Supplementary Figures 1 and 2 and the following sentence within the first paragraph of the Supplementary Methods:

'The surgical targets investigated here were the dorsal motor region of the STN and the posterior third of the ventral pallidum in the two cohorts.'

Some variability in the localisation of contacts between patients is inevitable and may relate to differences in individual anatomy. Importantly however, the visual pattern of contact location variability in the present report is not dissimilar to that seen in other cohorts of STN and GPi patients^{10,18} (see also Figure 2 in https://www.researchgate.net/publication/349008356_Effective_subthalamic_and_pallidal_deep_brain_stimulation_-_are_we_modulating_the_same_network).

An important point is that some localisation variability is beneficial as it introduces variance into measures that may be anatomically dependent such as hyperdirect pathway tract densities and coherence. It is partly this variance that is crucial in allowing us to determine relationships between such measures.

5) The authors presume that the hyperdirect pathway is exaggerated in PD. They should also acknowledge the literature that is contrary to this idea (e.g. Chu et al. 2017 <https://doi.org/10.1016/j.neuron.2017.08.038>, Matthai et al. 2015 <https://doi.org/10.1093/brain/awv018>). The authors should probably temper the claim made in their title “Neural signatures of pathological hyperdirect pathway activity in Parkinson’s disease”. It is not apparent that the authors present data that shows that the high beta cortico-STN coherence observed is in fact pathological, particularly given that recordings were made on medication.

We agree with the reviewer’s points. We have now changed our manuscript title to, ‘*Neural signatures of hyperdirect pathway activity in Parkinson’s disease*’.

We have also now discussed the two references pointed out by the reviewer which suggest that loss of hyperdirect pathway integrity may also be important in PD. Please see the final paragraph under the heading, ‘*Insights from the computational modelling*’ in the revised Discussion:

*‘Finally it is worth mentioning that although the data in this report are consistent with a role of an exaggerated hyperdirect pathway in the generation of abnormal synchrony in PD, other reports in monkeys and rodents suggest that a loss of hyperdirect pathway inputs to the STN can also reproduce the PD phenotype^{54,55}. Interestingly, this finding can be reconciled with our model when examining **Figures 8A and 8B**. If we assume that in the healthy state hyperdirect pathway strength is high, (e.g. $W_{ES} > 12$) this will correspond to the stable regime of the STN-GPe loop in Figure 8B where high beta band activity is propagated from the cortex to the subcortex. Pathological decreases in hyperdirect pathway strength will result in the system approaching a bifurcation and then additionally producing low beta frequency oscillations.’*

Additionally we also cite work which like our own is suggestive of a role for an exaggerated hyperdirect pathway in PD. Please see the first paragraph under the heading, ‘*Insights from the computational modelling*’ in the revised Discussion:

‘The model presented here underscores the likely importance of an exaggerated hyperdirect pathway in PD, which has also been suggested in previous reports^{4,18,21,39}.’

6) Lines 72-79: While the statement line 74 regarding lower beta suppression and its relationship to clinical improvement is valid and important, it is unclear how those findings “lead to the hypothesis that cortical coupling with the STN at high beta

frequencies may reflect hyperdirect pathway activity”. Some rewording / reorganizing of these sentences may be appropriate.

We apologise for the lack of clarity here. The sentences in question have now been reorganised as suggested to read as follows (see 4th paragraph of revised Introduction):

‘Synchrony at low beta frequencies (13-21 Hz) is detectible in the parkinsonian STN and is considered to be pathological as it is suppressed by both DBS and L-dopa therapy²²⁻²⁴ with some studies also demonstrating a correlation between the extent of treatment related beta suppression and clinical improvement^{4,25-28}. Previous work also demonstrates that synchronous activity between the cortex and the STN predominates at higher beta frequencies (21-30 Hz) and is segregated such that mesial motor/premotor areas drive STN activity across this frequency range^{2,4}. This leads to the hypothesis that cortical coupling with the STN at high beta frequencies may reflect hyperdirect pathway activity⁴.’

7) Lines 79-81: “If high beta frequencies are reflective of the hyperdirect pathway, synchrony within this frequency range should be relatively restricted to the STN and its interactions with cortex.” High beta band activity has been observed outside of the STN, and high beta coherence has been observed between cortex and GPi (Williams et al. 2002, <https://doi.org/10.1093/brain/awf156>, see also Talakoub et al. 2016, <https://doi-org.ezp3.lib.umn.edu/10.1038/srep34930>). Given these and other studies, it would seem that synchrony in this frequency range is not restricted to the STN and cortex. At least authors should acknowledge some of the literature that already addresses and may not necessarily support this hypothesis (that high beta synchrony is mostly restricted to STN and cortex)

We thank the reviewer for pointing this out to us. We have now modified the statement in question and included the two references suggested by the reviewer as follows (see 4th paragraph of revised Introduction):

‘If high beta frequencies are reflective of the hyperdirect pathway, synchrony within this frequency range might be expected to be greater within the STN and its cortical network than within other basal ganglia structures such as the GPi. However, due to the connectivity between STN and GPi, any differences between these two sites are likely to be relative rather than absolute. Thus, we note that high beta activity has been detected in both the STN and the GPi^{3,29}. In addition to the relative differences in power, we would predict an overlap between cortico-STN anatomical connectivity of the hyperdirect pathway and the profile of cortico-STN functional connectivity at high beta frequencies.’

We very much agree with the reviewer that high beta peaks are observed outside the STN and within the GPi. For example spectral peaks at high beta frequencies are observed in our own data from both UCL and Shanghai in Figures 2, 3 and 4. Nevertheless, in both datasets we observe greater high beta band power and cortical coherence for the STN when compared to the GPi. These findings are also captured by our computational model, which also highlights that the differences in high beta are relative.

8) Lines 83-85: “...STN beta waveform features suggestive of synchronisation 29,30 to a cortical source...” While cited studies indicate waveform shape is suggestive of

synchronisation, can authors assert that it the source is necessarily cortical? Could it not reflect synchronous inputs from other connected sites (e.g. GPe, CM/PF).

We thank the reviewer for raising this issue. In the revised manuscript we decided to remove the sections covering waveform shape analysis as advised by Reviewer 3 (see below). We agree that that analysis of waveform non-sinusoidality at a single subcortical site is not necessarily suggestive of cortical synchronisation – despite the fact that this measure was correlated with hyperdirect pathway fibre tract densities. We are currently developing methods that will facilitate the analysis of how non-sinusoidality may be synchronised between the cortex and the STN. This analysis now however is beyond the scope of the present paper and will be reserved for a separate manuscript.

9) Line 118: MNI has not been previously defined

The acronym MNI has now been defined in the Methods section of the revised paper, under the heading ‘*DBS electrode localisation and fibre tracking*’ as follows:

‘In order to then compare electrode placement across subjects, pre- and postoperative MRI acquisitions were non-linearly co-registered (normalisation) into MNI ICBM152 NLIN 2009b stereotactic space (Montreal Neurological Institute; <https://www.bic.mni.mcgill.ca/ServicesAtlases/ICBM152NLin2009>) using the SPM12 segment nonlinear option in Lead-DBS.’

10) Figure 3 D-F; it is not readily apparent that the black arrows correspond to distinct peaks in the coherence plots. There is a broad flat peak in cortico-STN coherence; why is that particular point with black arrow highlighted? The low frequency arrow in D does not appear to be a distinct peak. Why does panel F lack an arrow?

We thank the reviewer for this feedback. We have now removed the arrows so as to avoid any lack of clarity. In all revised plots, which include new data (revised Figures 2, 3 and 4) there is in most cases a small peak at around 13 Hz which we define to be in the low beta frequency range. This is followed by a larger and broader peak at higher beta frequencies.

11) Rest recordings had a duration of 3 minutes. How stationary were the recordings across time (both PSD and coherence measures)? Is it feasible to divide the recording duration into smaller segments and average (therefore possibly less susceptible to brief epochs that might be outliers)?

Thank you for raising these important issues. Regarding the first question about stationarity, it is now increasingly believed that beta band activity within the basal ganglia is not stationary and that it instead organises into short lived bursts¹⁹⁻²¹. The question regarding the stationarity of interactions between the cortex and the STN/GPi in patients with PD remains less clear however. We are currently developing methods using Hidden Markov Models to look at the stationarity of cortico-STN interactions, which is outside the scope of the present manuscript and will be the subject of future work. Importantly our own model makes certain predictions in this regard which we hope to test. These are detailed in the Supplementary Results section under the heading, ‘*Comment on bursting behaviours*’:

*'Recent work has focussed on the importance of transient rather than sustained episodes of synchrony being important in both pathological and physiological states²⁶⁻²⁸. Although the model presented here does not explicitly focus on the generation of bursting behaviours, it is easy to see how bursts may arise. Endogenous fluctuations (triggered by noise) of the inputs to the reciprocal STN-GPe loop when it is operating close to its bifurcation point may trigger transient oscillatory behaviours (see the left image in **Figure 8B**). This model therefore makes the testable prediction that transient bursts of cortical high beta activity can trigger the generation of lower beta frequency bursts within the STN. The genesis of bursting activity within similar models is likely to be a focus of future work.'*

Regarding the reviewer's second question about averaging, we apologise for a lack of clarity in this regard. We did in fact compute spectra and cross spectra over trials each of a 3 second duration which were subsequently averaged. We have now added this information to the Supplementary Methods section, under the heading '*Simultaneous magnetoencephalography and local field potential recordings*':

'Rest recordings had a duration of 3 minutes and merged MEG-LFP data were epoched into trials of 3 second duration which were subsequently averaged after the computation of auto- and cross- spectral measures.'

12) The clinical characteristics between the STN and GPi cohorts appear quite different (e.g. UPDRS-III scores). Could differences in patient phenotype underlie the differences observed in high beta, rather than differences between STN and GPi connectivity? The authors discussion of this being a study limitation is appreciated, however the cohorts are receiving very different amounts of levodopa; again, if any data were collected off medication in these patient cohorts, analysis with those data would be very beneficial. Although on-medication data collection is mentioned as a limitation, it perhaps should not be dismissed saying high beta is unlikely to be impacted. Though referenced studies may show minimal impact in the STN, the literature on GPi DBS LFPs off and on medication is more sparse, and as already mentioned has been shown to impact high beta GPi coherence measures in previous studies.

Please refer to Response 3 to reviewer 1 for an in depth discussion of this important issue.

In summary there were indeed phenotypic differences between the STN and GPi groups for the UCL cohort. In the newly included Shanghai cohort however the groups are much more closely matched (see updated Supplementary Table 1). The finding that results from the UCL cohort closely match those from the Shanghai cohort indicates that phenotypic differences are unlikely to be responsible for the group differences observed.

Additionally, for the UCL cohort where phenotypic differences were observed we accounted for them in statistical analyses by including them as covariates. This is statistically equivalent to 'regressing out' the effects of variables for which there were phenotypic differences before testing for group differences. Finally in line with the reviewer's suggestion we have included on and off medication recordings for the UCL STN cohort (see also Response 2 for further discussion of medication effects). The fact that we observed no medication effects on high beta band power or coherence for the STN serves to provide some further evidence that levodopa dose differences and UPDRS-III score differences are unlikely to account for any group differences observed.

13) Line 166: “We characterised the directionality of cortico-subcortical coupling at beta frequencies in order to determine whether these signals originate from the cortex or subcortical structures.” Stronger Cortex->STN Granger causality measures do not necessarily imply that is due to the hyperdirect pathway, does it? Could it not be striatal in origin via the indirect pathway?

We thank the reviewer for raising this point. In the present report we observed that mesial cortical areas predominantly drive STN activity at high beta frequencies. It is true that activity from cortex could be propagated to the STN from two routes: 1) the hyperdirect pathway between the cortex and the STN and 2) the indirect pathway via the striatum.

Using dMRI we were able to isolate hyperdirect pathway fibres by excluding those not traversing the striatum on their passage between cortex and STN. We observed a strong relationship between hyperdirect pathway tract density and cortico-STN high beta band coherence which is suggestive of the hyperdirect pathway being a key route of transmission of high beta band activity.

As the reviewer points out, this does not however exclude the possibility that high beta activity is also transmitted via the indirect pathway. We have added an additional analysis in our revised manuscript in order to further probe this. Instead of studying fibre tracts bypassing the striatum we included only fibre tracts passing through the striatum on their passage to the STN or GPi as this maybe more reflective of indirect and direct connections to these two structures. Using this approach however we observed no relationships between fibre tract densities and coherence in the high and low beta bands. These results would therefore favour the notion that that the dominant route of transmission of high beta band activity to the STN occurs via the hyperdirect pathway.

We have updated our methods section (see under heading *‘relationship of structural and functional connectivity’*), describing the new analyses as follows:

‘Accordingly a striatal mask was created using the DISTAL atlas within Lead-DBS⁶⁶. In separate analysis we included only fibres passing through the striatum on their passage to the STN or GPi as these maybe representative of non-hyperdirect connections from the indirect and direct pathways respectively.’

The results section has also been updated to include the following, under the heading, *‘Functional connectivity is predicted by anatomical connectivity within the cortico-STN hyperdirect pathway’*:

‘In separate analysis, rather than limiting chosen fibres to those bypassing the striatum (indicative of hyperdirect connections), we selected fibres that passed through the striatum on their passage to either the STN or GPi. This procedure served to select fibres that may form part of the indirect and direct pathways to the STN and GPi respectively. Using this approach we observed no significant relationship between tract densities and coherence in either the high or low beta bands for both cohorts. Taken together, our findings suggest that cortico-STN coherence in the high beta band is related strongly to structural connectivity within the hyperdirect pathway.’

14) Do the time delays presented (20ms) support the argument that the hyperdirect pathway is driving this connectivity? I would point out the following statement from Williams et al. 2002:

“Where cortex drives STN this could be achieved either directly through the large cortico-subthalamic projection, or indirectly via the putamen/globus pallidus externa (Parent and Hazrati, 1995). A delay of ~20 ms would seem to favour the latter”

This is an important and interesting point which certainly warrants some further discussion. A number of studies have attempted to quantify cortical evoked potential latencies resulting from STN stimulation in patients undergoing DBS. In this regard the monosynaptic hyperdirect pathway has been identified by the observation of relatively short evoked potential latencies ranging from ~2-8ms²²⁻²⁵.

Importantly evoked response latencies are likely to be shorter than delays computed from phase based estimates for two potential reasons. Firstly, delays from phase based estimates depend not only on conduction delays but also on synaptic integration delays (in other words the delay for a neuronal population to synchronise its own activity to that of an input) which increase as more synapses are involved, and may be extended by inhibitory inputs to the STN. Secondly, in our own data we found a strong relationship between cortico-STN hyperdirect fibre densities and cortico-STN high beta band coherence. This same relationship was not observed when we considered fibres traversing the striatum (which are likely to be reflective of indirect pathway connections) on their passage from the cortex to the STN, suggesting that the hyperdirect pathway maybe the predominant route of transmission of high beta activity to the STN. Nevertheless it is possible that there will be some mixing of hyperdirect and indirect pathway components that overlap in frequency leading to an increased estimate of time delays between the cortex and the STN^{26,27}.

The above reasons may explain discrepancies in hyperdirect pathway evoked response latencies and cortico-STN delays estimated from cross-correlation measures in a recent paper published in *Neuron* by Chen and colleagues^{22,28}. In this report Chen et al. reported evoked response latencies of ~2ms within the hyperdirect pathway. Cross-correlation of evoked activity within the cortex and the STN however revealed that cortical activity led STN activity by ~60ms (see Figures 3D and 3E from Chen et al²²). More importantly the magnitude of cross-correlation at these longer latencies predicted stopping behaviours, highlighting that in vivo physiological information transmission within the hyperdirect pathway may occur more slowly than evoked response latencies. In view of these points an estimated delay of ~15-20 ms, as per our own findings for cortico-STN hyperdirect pathway transmission may not necessarily be excessive.

Additionally, it is worth also pointing out that we were especially interested in relative differences in transmission delays to the STN and GPi rather than in absolute values. Our finding of shorter delays between the cortex and STN than between cortex and GPi further supports the notion that high beta activity originates within cortex and is propagated to basal ganglia.

We have now summarised some of the above discussion points in our revised manuscript, in the discussion as follows:

'We estimated a delay of ~15-20 ms for cortico-STN hyperdirect transmission in the high beta band. Previous studies of cortical evoked responses to STN stimulation have identified the monosynaptic hyperdirect pathway based on the observation of response latencies ranging from 2-8 ms^{13,14,47,48}. Importantly evoked response latencies are likely to be shorter than delays computed from phase based estimates for a number of reasons. Firstly, delays from phase based estimates depend not only on conduction delays but also on synaptic integration delays (in other words the delay for a neuronal population to synchronise its own activity to that of an input) which increase as more synapses are involved, and may be extended by inhibitory inputs to the STN. Secondly, it is possible that there will be some mixing of hyperdirect and indirect pathway components that overlap in frequency leading to an increased estimate of time delays between the cortex and the STN^{4,49}. We found relatively little evidence of this in our own data however since we only observed a correlation between cortico-STN hyperdirect fibre density and cortico-STN high beta band coherence and not between cortico-striatal-STN fibre densities (indicative of indirect pathway connections) and cortico-STN high beta band coherence. These observations suggest that the hyperdirect pathway maybe the predominant route of transmission of high beta activity to the STN.

The above reasons may explain discrepancies in hyperdirect pathway evoked response latencies (which were ~2ms) and cortico-STN delays estimated from cross-correlation measures (which were ~60ms) in a recent paper published by Chen and colleagues^{11,50}. Importantly the magnitude of cross-correlation at these larger latencies predicted stopping behaviours, highlighting that in vivo information transmission within the hyperdirect pathway may occur significantly more slowly than evoked response latencies. Importantly we also observed relative differences in transmission delays from the cortex to the STN and GPi. The finding of longer delays to the GPi, which lies further downstream in the cortico-basal-ganglia circuit support the notion that high beta activity originates within cortex and is propagated to basal ganglia.'

15) Figure 5: To provide the reader with greater intuition about the finding “tract density predicts high beta coherence,” showing data across patients (e.g. scatterplot of fiber density vs high beta coherence across patients) might be helpful.

We are grateful for this suggestion. As suggested by the reviewer we have in our updated manuscript provided more intuitive figures displaying the R^2 correlation coefficient (proportion of variance explained) at each voxel within each significant cluster for the correlation between tract density and coherence (see revised Figures 6 and 7). This is also explained in the methods section, in the 2nd paragraph under the heading, 'Relationship of structural and functional connectivity' as follows:

'The F-statistics of voxels within each significant cluster and the corresponding R^2 correlation coefficient were then written to a 3D image for visualisation.'

See also the following updated sentences in the Results section, under the heading, 'Functional connectivity is predicted by anatomical connectivity within the cortico-STN hyperdirect pathway':

*' R^2 correlation coefficient maps for voxels within each significant cluster are displayed in **Figure 6B**.'*

And:

*'Electrode localisations for this cohort are displayed in **Figure 7A**, whilst **Figure 7B** displays the corresponding F statistics and R^2 correlation maps.'*

16) Lines 334-337: The hyperdirect cortico-STN connections are not the only difference in connectivity between STN and GPi; it is a leap to assume that “by comparing LFPs from patients with STN electrodes to those from patients with GPi electrodes, we were able to isolate electrophysiological markers of hyperdirect pathway activity.” This seems too strong a statement.

We apologise for our poor wording here. The main approach for isolating electrophysiological markers of the hyperdirect pathway was the integration of high resolution dMRI derived structural connectivity of the hyperdirect pathway (from both connectomes and individual patient data) with functional connectivity derived from simultaneous MEG and STN LFP recordings. Additionally we compared power and cortical coherence profiles of the STN and GPi, which revealed features suggestive of a relative lack of hyperdirect inputs to the latter. This has now been updated in the first paragraph of the Discussion as follows:

'We were able to isolate electrophysiological markers of hyperdirect pathway activity by integrating high resolution dMRI derived structural connectivity – from both connectomes and individual patient dMRI - with functional connectivity derived from simultaneous MEG and STN LFP recordings.'

17) Although there are many figures and the current manuscript is extensive, the results of this study may be stronger if it included data during a motor task (though perhaps this was not collected), which might engage the hyperdirect pathway more distinctly than at rest.

We thank the reviewer for raising this. We agree that our findings lead to testable predictions about the involvement of high beta band cortico-STN coherence in motor behaviours such as stopping. We do not have task data from the patients studied in the present manuscript, and as the reviewer suggests the study of task behaviours would be best reserved for a separate manuscript that builds on the hypotheses presented here. We are intending to perform task based studies in the future.

Supplementary Methods

18) Please define/describe MNI space

This has now been defined in the following revised sentence:

'The source space was defined as a 5 mm spaced grid in MNI (Montreal Neurological Institute) template space bounded by the inner skull surface.'

19) Line 94: should read duration ≥ 400 ms ?

We have now removed the section on waveform non-sinusoidality and therefore this no longer appears in the Supplementary Methods (see response to point 8 and also Reviewer 3

point 8).

20) What was the magnet strength on the MRI imaging?

We apologise for omitting this important information. The field strength of the immediate postoperative MRI at UCL was 1.5 T. We have updated the relevant sentence in the Supplementary methods to the following:

'At UCL, the locations of the electrodes were confirmed following implantation with immediate postoperative fast spin-echo T2-weighted magnetic resonance imaging (MRI 1.5 T) with a Leksell frame still in situ.'

We thank the reviewer for their comprehensive and thorough reading of our manuscript and for the helpful points that they raised.

Reviewer 3

The main hypothesis of this paper that cortical coupling with the STN at high beta frequencies reflects hyperdirect pathway activity, that synchrony within this frequency range should be restricted to the STN and its interactions with cortex, and the authors predict an overlap between cortico-STN anatomical connectivity of the hyperdirect pathway and the profile of cortico-STN functional connectivity at high beta frequencies. This is an interesting thesis that the authors have developed from previous work and may be very important in the field.

We thank the reviewer for their overall positive appraisal of our manuscript.

Despite the statistical methods to correct for the multiple variables that could contribute to the results it is difficult to judge the interpretation of the results due to several confounding variables that will affect the LFP recorded: the participants were all on medication, the lead and electrode pair location varied in both the STN and GPi, the cohort was small and the GPi group had been chosen for the NBM target and were cognitively impaired. If any or many of these factors altered the LFP recorded it would be subject specific and hard to correct for in my opinion.

Major concerns

1) Small heterogeneous cohort, 6 patients with GPi DBS and 6 patients with STN DBS are used in the study. There is no mention of N in the abstract, nor the number data points used for each result. There is a difference in cognitive scores, as the cohort used for GPi appear to be the cohort used for the NBM target for PD dementia.

We have taken these important points into consideration when revising our manuscript. By way of summary, we have now significantly increased our sample size by including a further 12 patients (6 with STN electrodes and 6 with GPi electrodes) with simultaneous MEG and LFP recordings from Shanghai that were more closely matched in terms of cognition and

medication doses. In addition, we have included a further 8 patients with simultaneous MEG and LFP recordings from UCL (see also response 1 to reviewer 1). We have been able to replicate our key findings in the independently collected patient dataset from Shanghai and in the revised cohort from UCL we find no medication effects on power or coherence spectra in the high beta band. In addition, in the revised cohort from UCL where there were phenotypic differences, we accounted for these in statistical analyses by including them as covariates. This is statistically equivalent to ‘regressing out’ the effects of variables for which there were phenotypic differences before testing for group differences. We have explained this in the updated Methods section as follows (please see third paragraph of Methods section which reads as follows):

‘Phenotypic differences (Age, Disease Duration, pre-operative Levodopa equivalent dose, pre-operative UPDRS Part III motor scores ON medication and the Mini-Mental State Examination(MMSE)) between STN and GPi DBS patients were compared using unpaired t-tests separately for the UCL and Shanghai cohorts. For the UCL cohort, the disease durations of the STN and GPi DBS patients were similar in spite of the GPi DBS subgroup being on average older. Additionally the STN subgroup had a higher levodopa equivalent dose, lower UPDRS Part III motor scores ON medication and higher cognitive performance scores as measured by the MMSE (see Supplementary Table 1 for statistics of comparisons). In contrast for the Shanghai cohort, there were no significant group differences between STN and GPi DBS patients. In order to account for phenotypic differences for the UCL cohort, we included the four clinical features which were significantly different between the groups as covariates in all further statistical analyses.’

Additionally we have updated the discussion of group differences in the Discussion as follows (see section titled ‘Study Limitations’):

‘Our findings should be considered in light of the following limitations. Firstly, we observed phenotypic differences between the STN and GPi DBS patient groups for the UCL cohort but not for the Shanghai cohort (see Methods). We accounted for phenotypic differences at UCL by including them as covariates in our statistical analyses. Importantly the fact that the results of STN vs. GPi comparisons were consistent from both the Shanghai and UCL cohorts highlights that the relative differences in observed STN and GPi activity and coupling are unlikely to have been driven by phenotypic differences.’

The abstract has a tight word limit of only 150 words, so we have now included the total sample size in the final paragraph of the Introduction:

‘A total of 32 patients were recruited from two separate surgical centres – one in London and one in Shanghai - and synchrony profiles of the STN and GPi networks were compared.’

Additionally we have now provided individual data points for results in Figure 2-7. Additionally Supplementary Figures 1 and 2 show the electrode contacts located within the STN or GPi for each subject. These contacts were used for subsequent analyses. This is described now in the first paragraph of the Methods section:

‘Contacts traversing either the STN or the GPi are coloured in blue and only data from adjacent contact pairs where at least one contact traversed either the STN or GPi was used

for subsequent analysis. For example in the case that only the most inferior contacts 0 and 1 traversed the STN bipolar contact pairs 01 and 12 were used for subsequent analysis.'

2) Can the authors confirm that the relative lack of power in the high beta range in the GPi cohort was not influenced by cognitive impairment or the effect of medication?

We thank the reviewer for this important point, which has also been addressed in response 3 to reviewer 1 and also in response 12 to reviewer 2 – please see these responses for more details.

In summary there were indeed phenotypic differences between the STN and GPi groups for the UCL cohort. In the newly included Shanghai cohort however the groups are much more closely matched, without significant phenotypic differences (see updated Supplementary Table 1). The finding that results from the UCL cohort closely match those from the Shanghai cohort indicates that phenotypic differences are unlikely to be responsible for the group differences observed.

Additionally, for the UCL cohort where phenotypic differences were observed we accounted for them in statistical analyses by including them as covariates. This is statistically equivalent to 'regressing out' the effects of variables for which there were phenotypic differences before testing for group differences. Furthermore we have also now included on and off medication recordings for the UCL STN cohort (see also response 2 to reviewer 2 for further discussion of medication effects). The fact that we observed no medication effects on high beta band power or coherence for the STN, in keeping with previous findings^{29,30}, serves to provide some further evidence that levodopa dose differences and UPDRS-III score differences are unlikely to account for any group differences observed. A similar relative lack of levodopamine effects on high beta band power within the GPi has also been noted¹⁵.

3) Another concern is that the study was done on medication and the LEDD varied among people and between targets: LEDD= 1149mg for the STN group and 647mg for the GPi group. The report that it was not significant attests to the variation among subjects. This will result in attenuation of beta power that would be heterogenic but not related to whether it was GPi or STN. The time course of the recording after a dose of medication will also be different and did not seem to be standardized. This will affect the LFP and makes the results hard to interpret, and it would not be corrected just by accounting for LEDD as a covariate as the degree of attenuation of beta power is not linear and is very subject and timing specific. Please clarify whether the UPDRS III and LEDD data were the pre-operative values or at the time of recording and how many days post lead implantation the studies were done. Was this different regarding the potential effect of the lesion effect on LFP power?

Thank you for raising these important points. These issues are also addressed in detail in responses 2 and 12 to reviewer 2.

With respect to the influence of medication status upon our results we have addressed this issue in three complimentary ways. As the reviewer points out, there were indeed differences in levodopamine equivalent doses between the STN and GPi groups for the UCL cohort. We have now included a further 12 patients from Shanghai, for which the STN and GPi groups are much more closely matched in terms of their medication dosing (see updated Supplementary Table 1). The finding that results from the UCL cohort closely match those

from the Shanghai cohort indicates that differences in medication dosing are unlikely to be responsible for the group differences observed.

Secondly in our revised UCL cohort we have included patients with both ON and OFF medication recordings. The fact that we observed no medication effects on high beta band power or coherence for the STN serves to provide some further evidence that levodopa dose differences are unlikely to account for any group differences observed.

Please see the following in the revised Results under the heading, '*Differences in local synchrony between the STN and GPi and medication effects on the STN at beta frequencies*':

'For the Shanghai STN and GPi cohort and for the UCL GPi cohort recordings were only performed ON medication. In contrast, for the UCL STN cohort in which patients underwent both ON and OFF medication recordings, we observed a region within low beta frequencies (yellow line in Figure 2B; peak $t = 4.24$, FWE $p = 3 \times 10^{-3}$) where low beta power was increased in the OFF state compared to the ON state. Importantly there were no medication effects on high beta power within the STN.'

Additionally for the STN UCL cohort there were no effects of medication on cortico-STN coherence, directionality or on the relationship between tract density and high beta band coherence. Please see the following statements in our revised manuscript:

Firstly, in the Results section under the heading, '*Cortical-subcortical coherence at high beta frequencies occurs preferentially within the cortico-STN network*':

'Finally for the STN UCL cohort comparison of cortico-STN coherence profiles ON and OFF medication revealed only a main effect of band, such that mesial motor areas including SMA were preferentially coupled to the STN at high rather than at low beta frequencies (Figure 3C; peak $t = 5.24$, FWE $p = 4 \times 10^{-3}$, at MNI co-ordinates 16 12 70). We observed no significant main effect or interaction of medication state.'

Secondly, please refer to the following in the results section, under the heading, '*Functional connectivity is predicted by anatomical connectivity within the cortico-STN hyperdirect pathway*':

'The results of cluster based permutation testing are shown in Figures 6A, and reveal a lateralised cluster encompassing the SMA (blue contour), where tract density was predictive of high beta band coherence for the UCL dataset both ON and OFF medication and for the Shanghai dataset.'

Thirdly, for the UCL cohort where phenotypic differences were observed, we accounted for them in statistical analyses by including them as covariates. This is statistically equivalent to 'regressing out' the effects of variables for which there were phenotypic differences before testing for group differences (see also response 1).

Finally the analysis of medication effects is now summarised in the revised Discussion in the section titled, '*Medication effects on high and low beta band power and coherence*':

'For the UCL STN cohort, in keeping with previous reports, we observed no medication effects on coherence or on power within the high beta frequency range^{2,29,53,54}. Levodopa administration did however lead to a significant reduction in low beta band power within the

STN as previously described^{22,27,29}. Importantly the relationship between hyperdirect pathway tract density and high beta band cortico-STN coherence was maintained regardless of medication state. These observations suggest that levodopamine, in contrast to DBS, may not target the hyperdirect pathway, but rather that it could target subcortical mechanisms responsible for the transduction of high beta frequencies into lower frequencies.'

We also apologise for lack of clarity about the timing of recordings following medication. This has now been clarified in the following statement in the revised Methods, under the heading, 'Patients and experimental details':

'Recordings were performed whilst patients were seated at rest either after overnight withdrawal of usual dopaminergic medication (OFF state) or approximately one hour after medication administration (ON state). Patients were examined by a movement disorders neurologist (who was present for the duration of the recordings) prior to recordings in order to ensure that they were in their usual on or off states.'

Additionally, please see the following statement within the second paragraph of the Supplementary Methods, which clarifies the timing of recordings after surgery:

'Recordings were performed between days 3–6 after electrode implantation.'

We have now also clarified that UPDRS part III motor scores and Levodopa equivalent doses were calculated pre-operatively. Please see Supplementary Table 1 and also the following statement at the start of the third paragraph in the Methods section:

'Phenotypic differences (Age, Disease Duration, pre-operative Levodopa equivalent dose, pre-operative UPDRS Part III motor scores ON medication and the Mini-Mental State Examination(MMSE)) between STN and GPi DBS patients were compared using unpaired t-tests separately for the UCL and Shanghai cohorts.'

We appreciate the reviewer's point about microlesional effects on LFP power. We have now made reference to this in the section titled, 'Study Limitations' within the discussion:

'Finally, our recordings were performed a few days after electrode implantation. It is known that the insertion of DBS electrodes can result in transient amelioration of parkinsonism and may influence LFP activity before stimulation has been started⁶⁰⁻⁶². This may be due to the physiological effects of the lesion (a so called 'stun effect') and also to placebo mechanisms⁶³. Nevertheless, studies of LFP activity months or years after initial implantation reveal strong similarities in beta activity and in relationships between this activity and clinical state in the early postoperative and delayed postoperative periods⁶⁴⁻⁶⁷.'

4) It is unclear for many of the results what data are being included. Based on the methods (e.g., ~lines 506-511 on pg 22 and ~ line 32 on pg 2 of the supplemental data), it appears that multiple channel pairs from the same hemisphere are being used for some participants. If one STN had multiple electrode pairs used this could bias the finding of high beta power, despite correction, especially given the small sample of 6 patients per group for any conclusions that are being drawn. For instance the high beta power in the STN spectra in Fig 2B appear to be driven by a few spectra and if these are all from the same nucleus this would bias the results. It would also be clearer to the reader if

individual data points were superimposed on the bar graphs and that it was clearly identified which and how many channel pairs were being used per nucleus.

We thank the reviewer for these important point. The reviewer is correct that we included data from between 0 and 3 bipolar channels for each hemisphere in each patient based on the localisation of individual contacts within each target nucleus. See the first paragraph of the results section which includes:

*‘Contact localisation for individual patients is shown in **Supplementary Figures 1 and 2**. Contacts traversing either the STN or the GPi are coloured in blue and only data from adjacent contact pairs where at least one contact traversed either the STN or GPi was used for subsequent analysis. For example in the case that only the most inferior contacts 0 and 1 traversed the STN bipolar contact pairs 01 and 12 were used for subsequent analysis.’*

For STN contacts as shown in Supplementary Figure 1, either 2 or 3 bipolar channels were used for every hemisphere. The chances of bias by a single hemisphere will therefore be very small since the number of bipolar channels that could be included from each hemisphere is small in comparison to the total number of bipolar channels. To highlight that the high beta spectral peaks observed in the STN were not driven by outliers we have included the following figure below which shows spectra for individual STN contact pairs for the UCL cohort recorded ON and OFF medication as per revised Figure 2B. Clear peaks for each trace falling within the high beta frequency range were identified (using the findpeaks algorithm in MATLAB) and marked with a red triangle. It is seen that a significant proportion of spectra display peaks within the high beta frequency range. These peaks are not easily visible to the naked eye without marking them since there are a large number of overlapping data points.

Furthermore, we have now increased our sample size by including data collected from a cohort of 6 STN and 6 GPi DBS patients recruited from the Ruijin University Hospital in Shanghai (see revised Figure 2). The spectral features recorded from the Shanghai STN cohort are strongly consistent with those recorded from the UCL STN cohort highlighting that the finding of high beta power within the STN is both robust and reproducible. Note that we have also reported high beta spectral peaks within the STN in previous reports^{27,29}.

Finally we have taken onboard the reviewer's suggestion and added individual data points to all bar graphs in Figures 5,6 and 7.

5) Was there any difference in high beta power between the STN and GPi data if the authors did not subtract the 1/f curve and can they provide evidence that this is a robust method that would not vary based on all of the variables above?

We thank the reviewer for this important question. We have in Figure 2 of our manuscript plotted both raw power spectra (left panels) and power spectra after normalisation with the FOOOF (fitting oscillations & one over f) algorithm (right panels). This algorithm is in our experience a robust approach which has been developed by Bradley Voytek's group (<https://foeof-tools.github.io/foeof/>) and has been recently published in Nature Neuroscience. The paper in Nature Neuroscience motivates and details the technicalities of the method³¹ and we have also used this approach in other recent papers in our group³².

Our motivation for using FOOOF was as a normalisation method, since it allows the extraction of oscillatory activity of interest that lies on a background 1/f component which may vary across subjects, recording sessions and due to other factors. Application of this approach allows us to more accurately compare oscillatory activity of interest within different frequency bands³².

In order to make our use of this method clear we have modified the following sentences (and added the appropriate reference) in the Methods section, under the heading, '*Analysis of oscillatory synchrony within the cortico-STN/cortico-GPi circuit*':

'To make spectra comparable across subjects we used a spectral parameterisation algorithm (Fitting Oscillations & One-Over F algorithm, <https://github.com/foeof-tools/foeof>) to model the aperiodic (1/f) component⁶⁹. This was visualised in all cases for quality control and subsequently subtracted from the power spectrum in order to isolate the periodic oscillatory component of interest.'

6) Was the power in Fig 2 normalized so it could be compared across STNs/GPis?

Thank you for this point. The FOOOF method served as a normalisation technique as per our response to point 5. Power values were then log transformed prior to statistical comparison. Please see the following statement in the Methods section, under the heading, '*Analysis of oscillatory synchrony within the cortico-STN/cortico-GPi circuit*':

'To test for differences in the spectra of STN and GPi at each frequency, mean (across trials) log spectral timeseries were converted into 1D images, smoothed with a 2.5 Hz gaussian kernel and subjected to a t-test within SPM.'

7) Lead location variability. In addition to the small number of nuclei/subjects the varying locations of the DBS leads in the STN and the GPI also introduce variability in the LFP. The GPI placement was targeting the NBM and not sensorimotor GPI and this might alter the LFP spectra. The HDP and other afferent inputs are likely conserved anatomically and so recordings from a variety of locations may contribute to variability in results.

We have now significantly increased our sample size and include data from a further 20 patients leading to a total sample size of 32 patients. For the UCL GPI cohort the bottom most contact, 0 was targeted to the NBM, whilst upper contacts were targeted to the sensorimotor GPI. Note that it is those upper contacts targeting the GPI that are considered in the present report. We have made this clear by revising Supplementary Figure 2 and by highlighting contacts located within the GPI in blue. Additionally please see the following statement in the Supplementary Methods, under the heading, ‘*Surgical Procedure*’:

‘The surgical targets investigated here were the dorsal motor region of the STN and the posterior third of the ventral pallidum in the two cohorts.’

Our revised Figure 1 and Supplementary Figures 1 and 2 highlight that these regions were well targeted in the majority of cases.

Although we have increased our sample size, which leads to greater sampling within the range of variability of electrode localisations, we acknowledge that some variability remains (see also response 4 to Reviewer 2). As the reviewer alludes to this also has a crucial beneficial effect in introducing variance into measures that may be anatomically dependent such as hyperdirect pathway tract densities and coherence. Without such variance we would not be able to test for the existence of robust relationships between coherence and tract densities.

8) Latency of the putative HDP. This perhaps may have been part of a puzzling result regarding monosynaptic latencies. Does it seem concerning to the authors that the observed time-delay between the STN and cortical sources is so long (>15 ms)? This is not on the time-scale that would be expected for a monosynaptic connection such as the hyperdirect pathway. Cortical evoked potentials following stimulation of the STN were recently shown to take only ~2 ms (see Chen et al., 2020 “Prefrontal-Subthalamic Hyperdirect Pathway Modulates Movement Inhibition in Humans”). With the functional connectivity analyses, there is no way to know if there is a 3rd (or more) region mediating the connection and the observed time-delay seems to highlight that concern.

We thank you for these important and very interesting points. The same questions were also raised by reviewer 2 and have been addressed in responses 13 and 14 to reviewer 2. For sake of clarity we have copied some of the key points from these responses below.

A number of studies have attempted to quantify cortical evoked potential latencies resulting from STN stimulation in patients undergoing DBS. In this regard the monosynaptic hyperdirect pathway has been identified by the observation of relatively short evoked potential latencies ranging from ~2-8ms²²⁻²⁵.

Importantly evoked response latencies are likely to be shorter than delays computed from phase based estimates for two potential reasons. Firstly, delays from phase based estimates depend not only on conduction delays but also on synaptic integration delays (in other words the delay for a neuronal population to synchronise its own activity to that of an input) which increase as more synapses are involved, and may be extended by inhibitory inputs to the STN. Secondly, in our own data we found a strong relationship between cortico-STN hyperdirect fibre densities and cortico-STN high beta band coherence. This same relationship was not observed when we considered fibres traversing the striatum (which are likely to be reflective of indirect pathway connections) on their passage from the cortex to the STN, suggesting that the hyperdirect pathway maybe the predominant route of transmission of high beta activity to the STN. Nevertheless it is possible that there will be some mixing of hyperdirect and indirect pathway components that overlap in frequency leading to an increased estimate of time delays between the cortex and the STN ^{26,27}.

The above reasons may explain discrepancies in hyperdirect pathway evoked response latencies and cortico-STN delays estimated from cross-correlation measures in a recent paper published in *Neuron* by Chen and colleagues ^{22,28}. In this report Chen et al. reported evoked response latencies of ~2ms within the hyperdirect pathway. Cross-correlation of evoked activity within the cortex and the STN however revealed that cortical activity led STN activity by ~60ms (see Figures 3D and 3E from Chen et al ²²). More importantly the magnitude of cross-correlation at these longer latencies predicted stopping behaviours, highlighting that in vivo physiological information transmission within the hyperdirect pathway may occur more slowly than evoked response latencies. In view of these points an estimated delay of ~15-20 ms, as per our own findings for cortico-STN hyperdirect pathway transmission may not necessarily be excessive.

Additionally, it is worth also pointing out that we were especially interested in relative differences in transmission delays to the STN and GPi rather than in absolute values. Our finding of shorter delays between the cortex and STN than between cortex and GPi further supports the notion that high beta activity originates within cortex and is propagated to basal ganglia.

We have now summarised some of the above discussion points in our revised manuscript, in the discussion as follows:

'We estimated a delay of ~15-20 ms for cortico-STN hyperdirect transmission in the high beta band. Previous studies of cortical evoked responses to STN stimulation have identified the monosynaptic hyperdirect pathway based on the observation of response latencies ranging from 2-8 ms ^{13,14,47,48}. Importantly evoked response latencies are likely to be shorter than delays computed from phase based estimates for a number of reasons. Firstly, delays from phase based estimates depend not only on conduction delays but also on synaptic integration delays (in other words the delay for a neuronal population to synchronise its own activity to that of an input) which increase as more synapses are involved, and may be extended by inhibitory inputs to the STN. Secondly, it is possible that there will be some mixing of hyperdirect and indirect pathway components that overlap in frequency leading to an increased estimate of time delays between the cortex and the STN ^{4,49}. We found relatively little evidence of this in our own data however since we only observed a correlation between cortico-STN hyperdirect fibre density and cortico-STN high beta band coherence and not between cortico-striatal-STN fibre densities (indicative of indirect pathway connections) and

cortico-STN high beta band coherence. These observations suggest that the hyperdirect pathway maybe the predominant route of transmission of high beta activity to the STN.

The above reasons may explain discrepancies in hyperdirect pathway evoked response latencies (which were ~2ms) and cortico-STN delays estimated from cross-correlation measures (which were ~60ms) in a recent paper published by Chen and colleagues^{11,50}. Importantly the magnitude of cross-correlation at these larger latencies predicted stopping behaviours, highlighting that in vivo information transmission within the hyperdirect pathway may occur significantly more slowly than evoked response latencies. Importantly we also observed relative differences in transmission delays from the cortex to the STN and GPi. The finding of longer delays to the GPi, which lies further downstream in the cortico-basal-ganglia circuit support the notion that high beta activity originates within cortex and is propagated to basal ganglia.'

9) Non-linear waveform interpretation and place in ms

It is slightly unclear what the purpose of the investigation of the nonlinear waveform features of the two regions is within the current experimental framework. These results are mentioned in the abstract and get a substantial section in the discussion, but then are actually only placed in the supplementary materials. Although there is certainly a growing interest in investigating the sharpness of beta oscillations (especially in PD), the authors argue that the observed difference in sharpness is reflective of a difference in synchronization of input. The actual direct evidence for this is still extremely limited and it is a large jump to take the observed differences in sharpness observed in the STN vs. GPi to say this is evidence for a “dominance of a very direct pathway from cortex to STN”.

We have taken on board the reviewer's point and have decided to remove the sections on waveform non-sinusoidality from our manuscript. We agree that waveform non-sinusoidality is not necessarily suggestive of cortical synchronisation – although we did observe that this measure was correlated with hyperdirect pathway fibre tract densities. We are in the process of developing methods that will facilitate the analysis of how non-sinusoidality may be synchronised between the cortex and the STN. This analysis now however is beyond the scope of the present paper and will be reserved for a separate manuscript.

10) Figure 3 – can the authors clarify whether coherence would be affected by power? If there was greater high beta power in the STN than the GPi then would that influence the greater coherence between the cortex and STN compared to the cortex and GPi in high beta?

Thank you for this interesting point. The data in our manuscript implicates cortico-subcortical functional connectivity (coherence) over the high beta frequency range as the predominant driver of subcortical high beta frequency power. High beta band activity within focal motor regions including the SMA was shown by granger causality analysis to drive activity over this same frequency range within both the STN and the GPi. Please see revised Figure 5 and also the following statements in the Results section under the heading, '*Directionality of beta band cortical-subcortical coupling and estimation of transmission delays*':

'For both the UCL and Shanghai cohorts, the difference in Granger causality was significantly greater than zero in the direction of SMA and mesial M1 leading the STN for high but not for low beta frequencies (Figures 5A, B and D).'

‘Similarly cortical activity in SMA and mesial M1 led activity within the GPi selectively over the high but not low beta frequency range (Figure 5C and E).’

Additionally these same cortical areas display both greater fibre connectivity and greater high beta band coherence with the STN rather than with the GPi (see Figures 3, 4 and 6). For the case of STN, but not the GPi there was a strong relationship between hyperdirect pathway fibre densities and high beta band coherence.

These structural and functional connectivity differences can account for the differences in high beta power and coherence between the STN and the GPi as is demonstrated by simulations of our computational model. Please see panel A of Figure 8 where we simulate the effect of hyperdirect pathway strength on high beta band power within the STN/GPi. It is seen that increasing hyperdirect pathway strength leads to increased power within both structures, although power is relatively greater (at each strength) within the STN compared to the GPi. Please also see Figure 9 and also the revised Results section, under the heading, *‘Modelling the effects of cortico-subcortical connectivity on the frequency and coherence of oscillations’*:

‘For a range of values of W_{GIE} oscillatory peaks within the high and low beta frequency ranges are seen in the STN and GPi. As observed in our own data (Figures 2, 3 and 4), both high beta band power and cortical coherence were greater for the STN than for the GPi.’

11) line 286, could the authors list a range of the cortical spike activity (I assume they mean 650 spikes/s or more after the bifurcation?)

We apologise for any lack of clarity here. The system only displays oscillatory behaviour between an input range of between 330-670 spikes per second. Importantly this input spike rate is the product of the cortical spike rate and the hyperdirect pathway strength. Above this input range oscillatory behaviour ceases and the system returns to stability. To clarify this we have revised the sentences in question to the following:

‘To illustrate this more clearly, in the left image of Figure 8B is a bifurcation diagram for STN and GPe activities for a reduced model of only the reciprocal STN-GPe network, where both populations receive striatal input and the STN receives fixed (non-oscillatory) excitatory cortical inputs. The x-axis plots the cortical input which is the product of the hyperdirect pathway connection strength, W_{ES} and the firing rate of the cortical excitatory population. The system displays a stable fixed point (black lines) which transitions to an instability and oscillatory behaviour at cortical input values between approximately 330-670 spikes/second. Above a spike range of approximately 670 spikes/second oscillatory behaviour ceases as a stable fixed point returns.’

12) It is apparent how changing the feedback loop delay changes high and low beta, but how do the model results directly relate to coherence?

We used our model to simulate power and coherence spectra separately for the STN and GPi. In the Results section see the third paragraph under the heading, *‘Modelling the effects of cortico-subcortical connectivity on the frequency and coherence of oscillations’*:

‘Next, using our model we simulated power spectra of the cortex, STN and GPi and coherences between cortex-STN and cortex-GPi (Figure 9A).’

Our model was able to reproduce peaks in the low and high beta frequency bands in both coherence and power spectra. Furthermore, our model captured empirically observed differences in high beta band power and coherence between the STN and the GPi (see Figure 9). Finally, our model was also able to predict the correlation between the relationship observed between hyperdirect pathway tract density and high beta band cortico-STN coherence. These points have been highlighted in the Results under the heading, '*Modelling the effects of cortico-subcortical connectivity on the frequency and coherence of oscillations*'. Please see the following sentences:

'As observed in our own data (Figures 2, 3 and 4), both high beta band power and cortical coherence were greater for the STN than for the GPi.'

'Finally Figure 9B reveals that the computational model developed predicts a monotonically increasing relationship between hyperdirect pathway strength W_{ES} and cortico-STN coherence in the high beta frequency range (21-30 Hz).'

We are extremely grateful to the reviewer for their very helpful comments.

References

1. Litvak, V. *et al.* EEG and MEG data analysis in SPM8. *Comput. Intell. Neurosci.* **2011**, 852961 (2011).
2. Kilner, J. M., Kiebel, S. J. & Friston, K. J. Applications of random field theory to electrophysiology. *Neurosci. Lett.* **374**, 174–8 (2005).
3. Kilner, J. M. & Friston, K. J. Topological inference for EEG and MEG. *Ann. Appl. Stat.* **4**, 1272–1290 (2010).
4. Friston, K. J. *et al.* Statistical parametric maps in functional imaging: A general linear approach. *Hum. Brain Mapp.* **2**, 189–210 (1994).
5. Oostenveld, R., Fries, P., Maris, E. & Schoffelen, J.-M. FieldTrip: Open source software for advanced analysis of MEG, EEG, and invasive electrophysiological data. *Comput. Intell. Neurosci.* **2011**, 156869 (2011).
6. Maris, E. Statistical testing in electrophysiological studies. *Psychophysiology* **49**, 549–65 (2012).
7. Friston, K. Ten ironic rules for non-statistical reviewers. *NeuroImage* **61**, 1300–1310 (2012).
8. Statistical Parametric Mapping: The Analysis of Functional Brain Images - Google Books. Available at: https://books.google.co.uk/books?hl=en&lr=&id=G_qdEsDlkp0C&oi=fnd&pg=PP1&dq=info:IJKLLKMTWZoJ:scholar.google.com&ots=XnZHDBQ2VL&sig=Fpsv-F4y41IggDmH-yp-sJ7BVRQ&redir_esc=y#v=onepage&q&f=false. (Accessed: 25th January 2021)
9. Wang, Q. *et al.* Normative vs. patient-specific brain connectivity in deep brain stimulation. *Neuroimage* **224**, 117307 (2021).
10. Horn, A. *et al.* Connectivity Predicts deep brain stimulation outcome in Parkinson disease. *Ann. Neurol.* **82**, 67–78 (2017).
11. Al-Fatly, B. *et al.* Connectivity profile of thalamic deep brain stimulation to effectively treat essential tremor. *Brain* **142**, 3086–3098 (2019).
12. Li, N. *et al.* A unified connectomic target for deep brain stimulation in obsessive-

- compulsive disorder. *Nat. Commun.* **11**, 1–12 (2020).
13. Neumann, W.-J. *et al.* Functional segregation of basal ganglia pathways in Parkinson's disease. *Brain* **141**, 2655–2669 (2018).
 14. Williams, D. *et al.* Dopamine-dependent changes in the functional connectivity between basal ganglia and cerebral cortex in humans. *Brain* **125**, 1558–69 (2002).
 15. Lofredi, R. *et al.* Pallidal beta bursts in Parkinson's disease and dystonia. *Mov. Disord.* **34**, 420–424 (2019).
 16. Kiebel, S. J., Tallon-Baudry, C. & Friston, K. J. Parametric analysis of oscillatory activity as measured with EEG/MEG. *Hum. Brain Mapp.* **26**, 170–177 (2005).
 17. Rasch, D. & Guiard, V. *The robustness of parametric statistical methods.* *Psychology Science* **46**, (2004).
 18. Horn, A., Neumann, W.-J., Degen, K., Schneider, G.-H. & Kühn, A. A. Toward an electrophysiological “sweet spot” for deep brain stimulation in the subthalamic nucleus. *Hum. Brain Mapp.* **38**, 3377–3390 (2017).
 19. Tinkhauser, G. *et al.* Beta burst dynamics in Parkinson's disease OFF and ON dopaminergic medication. *Brain* **140**, 2968–2981 (2017).
 20. Tinkhauser, G. *et al.* The modulatory effect of adaptive deep brain stimulation on beta bursts in Parkinson's disease. *Brain* **140**, 1053–1067 (2017).
 21. Cagnan, H. *et al.* Temporal evolution of beta bursts in the parkinsonian cortical and basal ganglia network. *Proc. Natl. Acad. Sci. U. S. A.* **116**, 16095–16104 (2019).
 22. Chen, W. *et al.* Prefrontal-subthalamic hyperdirect pathway modulates movement inhibition in humans. *Neuron* 1–10 (2020). doi:10.1016/j.neuron.2020.02.012
 23. Miocinovic, S. *et al.* Cortical potentials evoked by subthalamic stimulation demonstrate a short latency hyperdirect pathway in humans. *J. Neurosci.* **38**, 9129–9141 (2018).
 24. Ashby, P. *et al.* Potentials recorded at the scalp by stimulation near the human subthalamic nucleus. *Clin. Neurophysiol.* **112**, 431–7 (2001).
 25. Devergnas, A. & Wichmann, T. Cortical potentials evoked by deep brain stimulation in the subthalamic area. *Front. Syst. Neurosci.* **5**, 30 (2011).
 26. Cassidy, M. & Brown, P. Spectral phase estimates in the setting of multidirectional coupling. *J. Neurosci. Methods* **127**, 95–103 (2003).
 27. Oswal, A. *et al.* Deep brain stimulation modulates synchrony within spatially and spectrally distinct resting state networks in Parkinson's disease. *Brain* **139**, 1482–1496 (2016).
 28. Narayanan, N. S., Wessel, J. R. & Greenlee, J. D. W. The Fastest Way to Stop: Inhibitory Control and IFG-STN Hyperdirect Connectivity. *Neuron* **106**, 549–551 (2020).
 29. Litvak, V. *et al.* Resting oscillatory cortico-subthalamic connectivity in patients with Parkinson's disease. *Brain* **134**, 359–74 (2011).
 30. López-Azcárate, J. *et al.* Coupling between beta and high-frequency activity in the human subthalamic nucleus may be a pathophysiological mechanism in Parkinson's disease. *J. Neurosci.* **30**, 6667–77 (2010).
 31. Donoghue, T. *et al.* Parameterizing neural power spectra into periodic and aperiodic components. *Nat. Neurosci.* **23**, 1655–1665 (2020).
 32. Gratwicke, J. *et al.* Resting state activity and connectivity of the nucleus basalis of Meynert and globus pallidus in Lewy body dementia and Parkinson's disease dementia. *Neuroimage* 117184 (2020). doi:10.1016/J.NEUROIMAGE.2020.117184

REVIEWERS' COMMENTS

Reviewer #1 (Remarks to the Author):

The revised work by Ashwini Oswal et al., which is now entitled „Neural signatures of hyperdirect pathway activity in Parkinson’s disease“ is a substantially revised version of the previously submitted work. The authors made a huge effort by including much more data. To this end, the authors partnered with a research group at Ruijin hospital, which is affiliated with the Jiao Tong University Shanghai. As a result, claims made in the paper are now based on a much broader data base. Doing this, the major shortcoming of this study has been overcome. Also, all my minor concerns have been addressed adequately.

Additionally, the statistical approach is much clearer and sufficiently justified now and the results are presented in a clearer and more complete way. Moreover, the discussion of the work is more detailed and balanced in the revised version and limitations are clearly addressed.

All in all, the authors made a huge effort to improve their work and address all issues raised by their three reviewers. I find the current version of the manuscript substantially more appealing and convincing. I would consider this manuscript as an interesting, thoughtful, and useful addition to the state-of-the-art, which will gain broad interest in its field of research.

Reviewer #2 (Remarks to the Author):

The authors have adequately responded to reviewer comments.

Reviewer #3 (Remarks to the Author):

The authors have satisfactorily answered all of my questions and I want to congratulate them on the effort to do so for all the reviewers comments.